# $d$-LINEAR GENERATION ERROR BOUND FOR DISTRIBUTED DIFFUSION MODELS

## ABSTRACT

The recent rise of distributed diffusion models has been driven by the explosive growth of data and the increasing demand for data generation. However, distributed diffusion models face unique challenges in resource-constrained environments. Existing approaches lack theoretical support, particularly with respect to generation error in such settings. In this paper, we are the first to derive the generation error bound for distributed diffusion models with arbitrary pruning, not assuming perfect score approximation. By analyzing the convergence of the score estimation model trained with arbitrary pruning in a distributed manner, we highlight the impact of complex factors such as model evolution dynamics and arbitrary pruning on the generation performance. This theoretical generation error bound is linear in the data dimension $d$, aligning with state-of-the-art results in the single-worker paradigm.

## 1 INTRODUCTION

Recently, distributed diffusion models have gained significant attention due to the explosive growth of data and the growing interest in data generation (Vora et al., 2024; de Goede et al., 2024). Specifically, in federated settings, diffusion models are trained collaboratively across multiple workers without the need to share personal sensitive data, such as images and audio, directly. This distributed approach enables large-scale data generation while avoiding the privacy risks and practical costs of centralizing data (Tun et al., 2023).

In real-world scenarios, workers typically possess limited computational and communication resources, which significantly hinder the performance (Zhang et al., 2021). When training a parametrized neural network in the reverse process of diffusion models, it would be unaffordable for resource-constrained workers to operate model updates. Some efforts have been made to address this challenge. For example, Li et al. (2024) propose DistriFusion, a method that divides the model input into multiple patches, each assigned to a GPU. This approach tackles the high computational costs involved in generating high-resolution images with diffusion models. Additionally, Lai et al. (2024) introduce an on-demand quantized energy-efficient distributed approach for training diffusion-based models in mobile edge networks. Despite the efforts made by these studies, they primarily focus on improving empirical performance, leaving the theoretical behavior as an open problem.

The theoretical lack in the generation performance of distributed diffusion models is driven by the increased complexity in training dynamics resulting from limited resources. The generation error bound of the diffusion model heavily depends on the loss value of the neural network trained in the reverse process (Benton et al., 2024; Chen et al., 2023). And the coupling between the loss and the gradient often reflects the convergence rate of the neural network (Zhou et al., 2024). As a result, accurately bounding the generation error requires describing the convergence rate of the parameterized neural network trained in the reverse process. However, resource limitations in distributed systems may lead to insufficient training or incomplete transmission of local models (Zhou et al., 2024; Qiao et al., 2023), which exacerbates global error accumulation and complicates the analysis of convergence rates during the reverse process.

In this paper, we provide formal theoretical support for distributed diffusion models in resource-constrained scenarios. To address the performance degradation caused by resource constraints, we

consider introducing pruning operations when training score estimation model in a distributed manner during the reverse process, and using coordinate-aware model aggregation (Zhou et al., 2024) to reduce global error accumulation. To obtain the convergence rate during the reverse process, we utilize the smoothness assumption to measure the inconsistency between the local and global gradients. We also implement a refined treatment of pruning errors and utilize the relationship between iterative model updates to explore their cumulative entanglement. By analyzing the convergence of the score estimation model and exploring the error between local and global training losses, we reveal the impact of complex factors such as the number of communication rounds and the number of workers on the local score estimation error. Using this actual error, rather than the assumed constant error in the single-worker paradigm (Benton et al., 2024; Chen et al., 2023; 2022), we derive the generation error bound for distributed learning diffusion models in resource-constrained scenarios. Specifically, our main contributions can be summarized as follows:

- **To the best of our knowledge, we are the first to incorporate the distributed learning dynamic of the score estimation model during the reverse process into the analysis of the final generation error.** We theoretically assess the discrepancy between the generated sample distribution and the actual distribution for each worker using KL divergence. This generation error bound aligns with the best-known results in the single-worker paradigm (Benton et al., 2024), exhibiting a linear dependence on the data dimension $d$. Notably, our framework can be seamlessly integrated with the theoretical error bounds of any diffusion model based on the single-worker paradigm under the perfect fractional approximation assumption. This integration ensures that the theoretical error bounds of similar distributed training architectures progress in tandem with advancements in the theoretical error bounds of the single-worker paradigm.

- We also derive convergence bounds for distributed learning of the score estimation model under arbitrary pruning, without relying on the bounded gradient assumption. It shows that the average gradient norm can converge at a rate of $\mathcal{O}(\frac{1}{\sqrt{\Gamma^* SQ}})$, showing the critical roles of the number of local training steps $S$ and the minimum parallel training degree $\Gamma^*$ in enhancing convergence efficiency.

## 2 RELATED WORK

In recent research, diffusion models (Song et al., 2020) have garnered widespread attention due to their remarkable achievements across multiple fields, including computer vision (Harvey et al., 2022), natural language processing (Li et al., 2022), temporal data modeling (Tashiro et al., 2021), and multi-modal learning (Ramesh et al., 2022; Ho et al., 2022). Particularly, some studies highlight that diffusion models not only generate high-quality data but also surpass traditional Generative Adversarial Networks in terms of stability and generation efficiency (Dhariwal & Nichol, 2021).

Recent works have extensively explored the theoretical performance of diffusion models. This provides a robust theoretical foundation for refining the model architecture and optimizing the training process. Initial studies on the convergence of diffusion models often requires restrictive assumptions about the data distribution, such as adherence to a log-Sobolev inequality (Yang & Wibisono, 2022), or results in bounds that are either non-quantitative (Pidstrigach, 2022) or exponential (Block et al., 2020) with respect to the problem parameters. Subsequent research has made significant improvements. For instance, some studies have achieved polynomial convergence rates for diffusion models without restrictive assumptions on the data distribution. Specifically, Chen et al. (2022) obtain polynomial error bounds in total variation (TV) distance, assuming that the score function is Lipschitz. They employ the Girsanov change of measure framework to analyze the discrepancy between the true and approximate reverse processes. Further advances are made by the work (Chen et al., 2023), which develop the Girsanov methodology further and introduce two important theorems: Theorem 2.1 shows that the KL divergence is linear in the data dimension but requires that $\nabla \log q_t$ be Lipschitz; Theorem 2.2 demonstrates that, under an early-stopping setting and with any data distribution having a finite second moment, the error is quadratic in the data dimension. Moreover, Benton et al. (2024) further improve the results under the early-stopping setting described in the work (Chen et al., 2023), achieving the current state-of-the-art error bound that is linear in the data dimension without smoothness assumptions on the data distribution.

However, most current research on diffusion models focuses on a single worker, primarily enhancing empirical performance or exploring theoretical attributes. With the advent of the big data era, dis-

tributed training (Qiao et al., 2023; Yuan et al., 2024) is emerging as a new trend, offering potential solutions to the scalability challenges posed by increasing data volumes. In response, DistriFusion is introduced in the work (Li et al., 2024), a method designed to run diffusion models across multiple devices in parallel, significantly reducing the latency associated with generating individual samples without compromising the quality of the generated images. Despite its practical effectiveness, the theoretical performance of DistriFusion (Li et al., 2024) is not explored. Additionally, Zhao et al. (2023) proposes FedDDA, a data augmentation-based federated learning architecture that utilizes diffusion models to generate data conforming to the global class distribution, thereby alleviating the non-IID data problem. However, theoretical exploration of this approach is also lacking.

In summary, there is a theoretical gap in collaboratively training diffusion models with resource constrains. To the best of our knowledge, we are the first to derive both convergence rate of the collaboratively trained score estimation model and error evaluation of locally generated samples.

## 3 PRELIMINARIES

### 3.1 DIFFUSION MODELS

The initial phase of the diffusion model is designed to progressively transform the given data distribution $q_0$, into a known prior distribution. This is referred to as the forward process, and it can be described using the Ornstein-Uhlenbeck (OU) process via the stochastic differential equation (SDE) (Pedrotti et al., 2023; Benton et al., 2024):

$$\mathrm{d}X_t = -X_t\mathrm{d}t + \sqrt{2}\mathrm{d}B_t, \quad X_0 \sim q_0 \tag{1}$$

where $(B_t)_{t\in[0,T]}$ denotes a standard Brownian motion on $\mathbb{R}^d$. Equation (1) aligns with a methodology known as Denoising Diffusion Probabilistic Models (DDPMs) (Ho et al., 2020), and is also referred to as Variance Preserving SDE in (Song et al., 2020). The OU process is favored for its analytically tractable transition densities, and it holds that $X_t|X_0 \sim \mathcal{N}(X_0e^{-t}, (1 - e^{-2t})\boldsymbol{I}_d)$.

We use $q_t(X_t), t \in [0, T]$ to denote the marginals of the forward process and then the reverse process satisfies the SDE:

$$\mathrm{d}X_t = -\{X_t + 2\nabla\log q_t(X_t)\}\mathrm{d}t + \sqrt{2}\mathrm{d}\tilde{B}_t, \quad X_0 \sim q_0 \tag{2}$$

where $(\tilde{B}_t)_{t\in[0,T]}$ is another standard Brownian motion on $\mathbb{R}^d$. By inverting the time direction $t$ with $T - t$ and setting $X_t = Y_{T-t}$, the reverse process (2) can be transformed to a forward one:

$$\mathrm{d}Y_t = \{Y_t + 2\nabla\log q_{T-t}(Y_t)\}\mathrm{d}t + \sqrt{2}\mathrm{d}B_t^{'}, \quad Y_0 \sim q_T \tag{3}$$

where $(B_t^{'})_{t\in[0,T]}$ is the standard Brownian motion on $\mathbb{R}^d$. The process $(Y_t)_{t\in[0,T]}$ can thus generate samples from the distribution $q_0$ by sampling $Y_0 \sim q_T$.

Nevertheless, Benton et al. (2024) pointed out that practical simulation of (3) necessitates overcoming certain challenges, which we also consider in this paper:

**(1) Score function estimation:** Since the score function $\nabla q_t(X_t)$ is unavailable, it is necessary to learn an estimation $s_\theta(X_t, t)$ of it. Specifically, the goal is to minimize the following loss function:

$$\int_0^T \mathbb{E}_{q_t(X_t)}[\| \nabla\log q_t(X_t) - s_\theta(X_t, t) \|^2]dt \tag{4}$$

While direct computation of (4) poses challenges, numerous score matching techniques (Hyvärinen & Dayan, 2005; Vincent, 2011) offer equivalent objectives that are more tractable. Among these, denoising score matching (Vincent, 2011) is utilized in this paper. Typically, we parameterize the score function $s_\theta(X_t, t)$, where $s_\theta$ represents the score, using a neural network with a parameter vector $\theta \in \mathbb{R}^D$. To optimize these parameters, we minimize the loss function through traditional SGD method over $\theta$, effectively training the neural network to accurately estimate the score function based on the input data $X_t$ and time $t$.

**(2) Unknown distribution approximation:** Sampling from the distribution $q_T$ is challenging due to the inaccessibility of $q_T$. Instead, sampling from the standard Gaussian presents a feasible alternative, as the OU process converges exponentially quickly to the standard Gaussian (Bakry et al., 2014; Chen et al., 2023).

**(3) Time discretization:** Given that equation (3) characterizes a continuous-time process, practical simulation requires the time variable to be discretized. This involves dividing the continuous time into a sequence of discrete points $0 = t_0 < t_1 < t_2 < \cdots < t_K \leq T$. Subsequently, we can initiate the process by sampling $\hat{Y}_0$ from the standard Gaussian and then concentrate on solving the SDE (also known as the exponential integrator Zhang & Chen (2022); De Bortoli (2022); Chen et al. (2023)) for each interval $[t_k, t_{k+1}]$ and $k = 0, \cdots, K - 1$:

$$\mathrm{d}\hat{Y}_t = \{\hat{Y}_t + 2s_\theta(\hat{Y}_{t_k}, T - t_k)\}\mathrm{d}t + \sqrt{2}\mathrm{d}\hat{B}_t \tag{5}$$

where $(\hat{B}_t)_{t \in [0,T]}$ is a standard Brownian motion. It allows for the approximation of the continuous-time dynamics of the process within each discrete interval, facilitating the practical simulation of the model. And we denote the marginals of the process (5) by $p_t$s.

**(4) Early stopping requirement:** Instead of running (5) to approximate the initial data distribution $q_0$, we opt to approximate the distribution $q_\delta$ as an early-stopping measure (Song et al., 2020). This strategy is deemed acceptable because, for a sufficiently small $\delta$, the discrepancy between $q_0$ and $q_\delta$ remains minimal. It is employed due to the potential for $\nabla \log q_t$ to rapidly increase, or "explode", as time $t$ approaches zero in non-smooth data distributions.

### 3.2 DISTRIBUTED LEARNING WITH ARBITRARY PRUNING

In the distributed learning framework, we consider a setup involving $N$ workers and a central server. These workers jointly undertake the task of learning a unified global model characterized by the parameter $\theta$. The objective is to optimize the following function:

$$\min_{\theta \in \mathbb{R}^D} F(\theta) := \frac{1}{N} \sum_{n=1}^{N} \underbrace{\mathbb{E}_{\xi_n \sim \mathcal{D}_n}[f_n(\theta, \xi_n)]}_{:= F_n(\theta)} \tag{6}$$

where $F_n(\theta)$ is a loss function defined on the dataset $\mathcal{D}_n$ based on the worker-$n$ specified $f_n(\theta, \xi_n)$, and $\xi_n$ signifies a data point sampled from the dataset $\mathcal{D}_n$.

In this learning framework , each worker keeps its own local dataset and conducts training operations with arbitrary pruning locally (Zhou et al., 2024). Communication with the server is restricted to the exchange of size-reduced model parameters (or gradients). More specifically, for the $q$-th round of the process, each worker-$n$ performs model pruning $\theta_{q,n,0} = \theta_q \odot m_{q,n}$ after receiving the latest global model parameter $\theta_q \in \mathbb{R}^D$ from the server, where $m_{q,n} \in \{0,1\}^D$ is a local mask generated based on mask policy $P$. And then $S$ steps of local training is performed to update pruned model parameters. The update rule can be described as follows:

$$\theta_{q,n,s} = \theta_{q,n,s-1} - \eta \nabla f_n(\theta_{q,n,s-1}, \xi_{n,s-1}) \odot m_{q,n} \tag{7}$$

Here, $s$ ranges from 1 to $S$, with $\theta_{q,n,0}$ representing the starting parameter for each round of updates at worker-$n$, and $\eta$ denotes the local learning rate used for the updates.

Upon completion of a round of training (encompassing $S$ steps) by all workers, the server aggregates all local parameters to form a new global model for the forthcoming round:

$$\theta_{q+1}^{(i)} = \frac{1}{|N_q^{(i)}|} \sum_{n \in N_q^{(i)}} \theta_{q,n,S}, \qquad \text{for each coordinate } i = 1, 2, \cdots, D \tag{8}$$

where $N_q^{(i)} = \{n : m_{q,n}^i = 1\}$ and we denote $\Gamma^* = \min_{q,i} |N_q^{(i)}| \geq 1$.

## 4 DISTRIBUTED LEARNING OF DIFFUSION MODELS WITH ARBITRARY PRUNING

When considering training a diffusion model across multiple workers in a distributed manner, the first objective is to optimize the function described in equation (6) to obtain the score estimation $S_{\theta_Q}(\cdot)$. By employing denoising score matching (Vincent, 2011) (see Appendix B for details), the term $F_n(\theta)$ in equation (6) can be expressed as

$$F_n(\theta) = \sum_{k=0}^{K-1} \gamma_k \mathbb{E}_{X_{n,0}, q(Y_{n,t_k}|X_{n,0})}[\|\, s_\theta(Y_{n,t_k}, T - t_k) - \nabla \log q(Y_{n,t_k}|X_{n,0}) \,\|^2] \tag{9}$$

where $\gamma_k = t_{k+1} - t_k$ is the length of the $k$-th discretized time interval, and $q_{n,t}, t \in [0, T]$ denotes the marginal of the forward process of worker-$n$. Therefore, $X_{n,0}$ is sampled from $q_{n,0}$ by worker-$n$, and $Y_{n,t_k} = X_{n,T-t_k} \sim q_{n,T-k}$. However, due to the randomness in actual training, such as sampling and noise randomness, we use $f_n(\theta, \xi_n)$ to represent the local loss with randomness during training. Specially, we assume the unbiasedness of $f_n(\theta, \xi_n)$, which is common in distributed scenarios, meaning that $\mathbb{E}[f_n(\theta, \xi_n)] = F_n(\theta)$.

As described in Section 3.2, after completing $Q$ rounds of distributed training (each with $S$ steps), we obtain the score function estimation $s_{\theta_Q}(\cdot)$. Starting from a pure noise state, the noise is gradually transformed into a form that approximates the original data. For each worker $n$, this process follows Equation (10), which incorporates the worker indicator into Equation (5).

$$\mathrm{d}\widetilde{Y}_{n,t} = \{\widetilde{Y}_{n,t} + 2s_{\theta_Q}(\widetilde{Y}_{n,t_k}, T - t_k)\}\mathrm{d}t + \sqrt{2}\mathrm{d}\widetilde{B}_{n,t} \tag{10}$$

where $(\widetilde{B}_{n,t})_{t \in [0,T]}$ is a standard Brownian motion, and we denote the marginals of the process (10) by $p_{n,t}$s. Specifically, the equation (10) can be solved explicitly by

$$\widetilde{Y}_{n,t_{k+1}} = e^{\gamma_k}\widetilde{Y}_{n,t_k} + 2[e^{\gamma_k} - 1]s_{\theta_Q}(\widetilde{Y}_{n,t_k}, T - t_k) + \sqrt{e^{2\gamma_k} - 1} \cdot \epsilon_{n,k}$$

where $\epsilon_{n,k} \sim \mathcal{N}(\mathbf{0}, \boldsymbol{I}_d)$. Further details can be found in Appendix C.

To obtain the main theoretical results, we rely on the following core assumptions.

**Assumption 1** *(**Lipschitzian gradient**). Loss function $F_n(\cdot)$s are with Lipschitzian gradients. i.e., For $\forall \theta, \phi \in \mathbb{R}^D$, it holds that*

$$\parallel \nabla F_n(\theta) - \nabla F_n(\phi) \parallel \leq L \parallel \theta - \phi \parallel$$

**Assumption 2** *(**Pruning-induced Error**). For an arbitrary mask $m_{n,q} \in \{0,1\}^D$ and an arbitrary model $\theta \in \mathbb{R}^D$, we assume that there exists $w^2 \in [0, 1)$:*

$$\parallel \theta - \theta \odot m_{n,q} \parallel^2 \leq w^2 \parallel \theta \parallel^2$$

**Assumption 3** *(**Bounded Variance**). For any model $\theta$ and sample $\xi$, there exist $\sigma_1 > 0$ and $\sigma_2 > 0$:*

$$\mathbb{E} \parallel \nabla f_n(\theta, \xi) - \nabla F_n(\theta) \parallel^2 \leq \sigma_1^2, \qquad \mathbb{E} \parallel \nabla F_n(\theta) - \nabla F(\theta) \parallel^2 \leq \sigma_2^2$$

**Assumption 4** *(**Data Distribution**). The data distribution $q_{n,0}$ of each worker-$n$ has finite second moments, and is normalized so that $\mathrm{Cov}(q_{n,0}) = \boldsymbol{I}_d$.*

These assumptions (Assumptions 1 to 4) are widely used in studies on diffusion models and distributed learning. Assumption 1 (Lian et al., 2017) is typically employed to ensure the stability and solvability of optimization problems, as it guarantees that the changes in gradients will not increase without bound. Assumption 2 (Zhou et al., 2024) guarantees that pruning operations do not degrade performance beyond a certain threshold, ensuring algorithm robustness. Assumption 3 Lian et al. (2017) restricts the influence of randomness on the optimization process. For Assumption 4, its first part ensures the convergence of the forward process, while the second part simplifies result descriptions, though it is not required for the analysis (Benton et al., 2024).

Building on Assumptions 1-3 mentioned above, we can establish the convergence bound for distributed learning of score estimation with arbitrary pruning.

**Theorem 1** *Under Assumptions 1-3, the following convergence result holds for distributed learning of score estimation with arbitrary pruning, provided that the step size $\eta$ satisfies $\eta \leq \min\{\frac{1}{27SL}, \sqrt{\frac{1}{3(8L^2S^2 + \frac{16S^2L^2w^2}{1-2w^2})}}, 1\}$ where $\Gamma^* = \min_{q,i} |N_q^{(i)}| \geq 1$, and pruning factor satisfies $w \in [0, \frac{\sqrt{2}}{2})$:*

$$\frac{1}{Q}\sum_{q=0}^{Q-1} \mathbb{E} \parallel \nabla F(\theta_q) \parallel^2$$

$$\leq \frac{6(F(\theta_0) - F(\theta_Q))}{\eta SQ} + (\frac{54\eta SL^3w^4}{Q(1 - 2w^2)} + \frac{18L^2w^4}{Q(1 - 2w^2)})\mathbb{E} \parallel \theta_0 \parallel^2 + \frac{9\eta LN\sigma_1^2}{(\Gamma^*)^2} +$$

$$(\frac{9\eta SL}{2} + \frac{3}{2})(\sigma_1^2 + \sigma_2^2)$$

Theorem 1 describes the rate at which the average gradient norm converges over all training rounds, which serves one of our main bounds. The term $\frac{6(F(\theta_0)-F(\theta_Q))}{\eta SQ}$ reflects the impact of iterative updates on the convergence behavior, while the remaining terms capture the combined effects of pruning operations, randomness, and local errors.

Specially, by tuning the appropriate step size $\eta$ in Theorem 1, we can directly derive the following result:

**Corollary 1** *Under Assumptions 1-3, if the step size $\eta$ satisfies $\eta = \sqrt{\frac{\Gamma^*}{SQ}}$, and pruning factor satisfies $w \in [0, \frac{\sqrt{2}}{2})$, and we can further set $Q \geq \max\{729\Gamma^*SL^2, \Gamma^*S, 3\Gamma^*(8SL^2 + \frac{16SL^2w^2}{1-2w^2})\}$ to further derive the convergence result of Theorem 1:*

$$\frac{1}{Q}\sum_{q=0}^{Q-1}\mathbb{E}\parallel \nabla F(\theta_q)\parallel^2$$

$$\leq \frac{6(F(\theta_0)-F(\theta_Q))}{\sqrt{\Gamma^*SQ}} + (\frac{2L^2w^4}{Q(1-2w^2)} + \frac{18L^2w^4}{Q(1-2w^2)})\mathbb{E}\parallel \theta_0 \parallel^2 + \frac{9LN\sigma_1^2}{\Gamma^*\sqrt{\Gamma^*SQ}} + \frac{5}{3}(\sigma_1^2 + \sigma_2^2)$$

Corollary 1 suggests that with an appropriately chosen step size $\eta$ and a sufficient number of training rounds $Q$, the convergence rate of distributed learning for score estimation with arbitrary pruning can be effectively dominated by $\mathcal{O}(\frac{1}{\sqrt{\Gamma^*SQ}})$. Increasing key hyperparameters—such as the number of training rounds $Q$, the number of local training steps $S$, and the minimum occurrences $\Gamma^*$ of any dimension parameter in the local model—results in tighter bounds on the average gradient norm. However, convergence can still be negatively affected by factors such as pruning-induced error, the gradient variance introduced by randomness, and discrepancies between local and global gradients.

When exploring the discrepancy between the distribution of the generated data and the true distribution of the original data, the following assumption is required for traditional single-worker architecture (Benton et al., 2024):

$$\sum_{k=0}^{K-1}\gamma_k\mathbb{E}\parallel s_\theta(Y_{n,t_k}, T-t_k) - \nabla\log q(Y_{n,t_k})\parallel^2 \leq \epsilon_{\text{score}}^2$$

However, this assumption may not fully capture the requirements of distributed diffusion model training, as it overlooks the complexity of training score estimation models in practice. By carefully addressing this, we clarify the influence of distributed training dynamics on the generated error bound, as detailed in Corollary 2.

**Corollary 2** *Suppose Assumptions 1-4 hold, $T \geq 1$, and there exists a constant $C > 0$, and some $\kappa > 0$ such that for each discretized time point $k = 0, \cdots, K-1$ we have $\gamma_k \leq \kappa \min\{1, T - t_{k+1}\}$. Then under the same settings of $\eta$ and $Q$ as in Corollary 1, for each worker-$n$, using the collaboratively learned model $\theta_Q$ aforementioned, it yields the following result when approximating the initial data distribution:*

$$KL(q_{n,\delta} \parallel p_{n,t_K})$$

$$= \mathcal{O}\big(F(\theta_0) + (\frac{\sqrt{\Gamma^*SQ}L^2w^4}{3Q(1-2w^2)} + \frac{3\sqrt{\Gamma^*SQ}L^2w^4}{Q(1-2w^2)})\mathbb{E}\parallel \theta_0 \parallel^2 + \frac{3LN\sigma_1^2}{2\Gamma^*} + \frac{5\sqrt{\Gamma^*SQ}}{18}(\sigma_1^2 + \sigma_2^2)$$

$$+ \parallel F_n(\theta_0) - F(\theta_0)\parallel +\sigma_2\parallel \theta_Q - \theta_0 \parallel +C(T-\delta) + \kappa dT + \kappa^2 dK + de^{-2T}\big)$$

In Corollary 2, the term $\parallel F_n(\theta_0) - F(\theta_0)\parallel +\sigma_2\parallel \theta_Q - \theta_0 \parallel$ captures the local-global error discrepancy. The term $C(T-\delta)$ arises from using denoising score matching to address the discretized form of Equation (4), while $\kappa dT + \kappa^2 dK$ is due to time discretization approximations, and $de^{-2T}$ governs the convergence of the forward process. The remaining terms are interpreted as the global loss associated with $\theta_Q$ which results from the distributed learning of score estimation with arbitrary pruning. Corollary 2 highlights how the training dynamics of the score estimation model affect the final generation error. This further demonstrates that the ideal constant error assumption on score approximation (Benton et al., 2024) is inadequate for practical distributed training scenarios.

**Remark 1** (*Suitable choice of Q, T and K*).*Consider the most extreme case when $\sigma_1^2 = \sigma_2^2 = 0$, which means that the target loss function of all workers is the same and the error caused by random sampling is negligible. We introduce $\epsilon^2$ to rewrite the KL error in Corollary 2 as $KL(q_{n,\delta} \parallel p_{n,t_K}) = \mathcal{O}(\epsilon^2 + (\frac{\sqrt{\Gamma^* SQ}L^2 w^4}{3Q(1-2w^2)} + \frac{3\sqrt{\Gamma^* SQ}L^2 w^4}{Q(1-2w^2)})\mathbb{E} \parallel \theta_0 \parallel^2 + \frac{3LN\sigma_1^2}{2\Gamma^*} + C(T-\delta) + \kappa dT + \kappa^2 dK + de^{-2T})$. At this point, for $T \geq 1$, $\delta < 1$, $K \geq \log(1/\delta)$, and some $\kappa = \Theta\left(\frac{T+\log(1/\delta)}{K}\right)$, if we set $Q = \Theta(\frac{\Gamma^* S}{\epsilon^4})$, $T = \Theta(\min\{\frac{1}{2}\log(\frac{d}{\epsilon^2}), \frac{\epsilon^2}{C}\})$ and $K = \Theta(\frac{d(T+\log(1/\delta))^2}{\epsilon^2})$, we have $KL(q_{n,\delta} \parallel p_{n,t_K}) = \mathcal{O}(\epsilon^2)$.*

## 5 THEORETICAL GUARANTEE

In this section, we outline the proofs of the main theoretical results, with a focus on Theorem 1 and Corollary 2.

### 5.1 PROOF SKETCH OF THEOREM 1

Utilizing the Lipschitzian gradient assumption, we start the proof by analyzing the change in the loss function during one round as the model transitions from $\theta_q$ to $\theta_{q+1}$:

$$\mathbb{E}[F(\theta_{q+1})] - \mathbb{E}[F(\theta_q)] \leq \underbrace{\mathbb{E}\langle \nabla F(\theta_q), \theta_{q+1} - \theta_q \rangle}_{B_1^{(q)}} + \underbrace{\frac{L}{2}\mathbb{E} \parallel \theta_{q+1} - \theta_q \parallel^2}_{B_2^{(q)}} \tag{11}$$

The first challenge in the theoretical analysis is bounding the terms $B_1^{(q)}$ and $B_2^{(q)}$. Based on the local update (7) and the global model aggregation (8), the key to analyzing these terms lies in measuring the inconsistency $B_3^{(q)}$ between the workers' and the server's gradients:

$$B_1^{(q)} \leq -\frac{S\eta}{2}\mathbb{E} \parallel \nabla F(\theta_q) \parallel^2 + \frac{1}{2S\eta}B_3^{(q)}$$

$$B_2^{(q)} \leq \frac{3NLS\eta^2\sigma_1^2}{2(\Gamma^*)^2} + \frac{3LS^2\eta^2}{2}\mathbb{E} \parallel \nabla F(\theta_q) \parallel^2 + \frac{3L}{2}B_3^{(q)}$$

$$B_3^{(q)} = \sum_{i=1}^{D}\mathbb{E} \parallel \frac{\eta}{|N_q^{(i)}|}\sum_{n \in N_q^{(i)}}\sum_{s=1}^{S}[\nabla F_n^{(i)}(\theta_{q,n,s-1}) - \nabla F_n^{(i)}(\theta_q)] \parallel^2$$

Since each worker trains on its own data, differences in local update direction naturally arise, and multiple local steps further exacerbate these discrepancies. Moreover, the arbitrary pruning operations of local models introduce dimensional inconsistencies in the submodels trained by different workers, necessitating a more refined analysis, which significantly increases the complexity.

**Measuring Inconsistency Between the Local and Global Gradients** Utilizing the Cauchy-Schwarz inequality and the Lipschitzian gradient assumption, we aim to transform the gradient deviation, represented by $B_3^{(q)}$, into a corresponding deviation in the model parameters:

$$B_3^{(q)} = \sum_{i=1}^{D}\mathbb{E} \parallel \frac{\eta}{|N_q^{(i)}|}\sum_{n \in N_q^{(i)}}\sum_{s=1}^{S}[\nabla F_n^{(i)}(\theta_{q,n,s-1}) - \nabla F_n^{(i)}(\theta_q)] \parallel^2$$

$$\leq \sum_{i=1}^{D}\frac{S\eta^2}{|N_q^{(i)}|}\sum_{n \in N_q^{(i)}}\sum_{s=1}^{S}\mathbb{E} \parallel \nabla F_n^{(i)}(\theta_{q,n,s-1}) - \nabla F_n^{(i)}(\theta_q) \parallel^2$$

$$\leq \eta^2 SL^2 \cdot \frac{1}{|N_q^{(i)}|}\sum_{n \in N_q^{(i)}}\sum_{s=1}^{S}\mathbb{E} \parallel \nabla\theta_{q,n,s-1} - \nabla\theta_q \parallel^2 \tag{12}$$

Note that the term $\mathbb{E} \parallel \theta_{q,n,s-1} - \theta_q \parallel^2$ above satisfies the following inequality:

$$\mathbb{E} \parallel \theta_{q,n,s-1} - \theta_q \parallel^2 = \mathbb{E} \parallel \theta_{q,n,s-1} - \theta_{q,n,0} + \theta_{q,n,0} - \theta_q \parallel^2$$

$$\leq \underbrace{2\mathbb{E} \parallel \theta_{q,n,s-1} - \theta_{q,n,0} \parallel^2}_{B_4^{(q)}} + \underbrace{2\mathbb{E} \parallel \theta_q \odot m_{n,q} - \theta_q \parallel^2}_{B_5^{(q)}} \tag{13}$$

The $B_4^{(q)}$ term reflects the model evolution caused by local multistep iterative training, while the $B_5^{(q)}$ term represents the error resulting from local arbitrary pruning. Collectively, these two terms lead to the difference between the local model $\theta_{q,n,s-1}$ at any step $s-1$ ($s = 1, \cdots, S$) and the global model $\theta_q$ at the beginning of the current round $q$.

**Exploring the Cumulative Entanglement of Arbitrary Pruning Operations and Local Multistep Training** Local multistep training causes the gradient to cumulatively affect the model update trajectory. Although the common bounded gradient assumption simplifies the analysis, it overlooks the cumulative impact of factors like random sampling noise. Relying solely on Assumption 2 to describe the pruning error neglects the model evolution dynamics, introducing additional non-deterministic dependencies in the final convergence result and making it less intuitive (Zhou et al., 2024). Based on the above considerations, we deal with $B_4^{(q)}$ and $B_5^{(q)}$ as follows:

$$B_4^{(q)} = 2\mathbb{E} \parallel -\eta \sum_{j=0}^{s-2} \nabla f_n(\theta_{q,n,j}, \xi_{n,j}) \odot m_{q,n} \parallel^2$$

$$\leq 2\eta^2(s-1) \sum_{j=1}^{s-1} \mathbb{E} \parallel \nabla f_n(\theta_{q,n,j-1}, \xi_{n,j-1}) - \nabla F_n(\theta_{q,n,j-1}) + \nabla F_n(\theta_{q,n,j-1}) - \nabla F_n(\theta_q)$$

$$+ \nabla F_n(\theta_q) - \nabla F(\theta_q) + \nabla F(\theta_q) \parallel^2$$

$$\leq 8\eta^2(s-1)^2(\sigma_1^2 + \sigma_2^2) + 8\eta^2 L^2(s-1) \sum_{j=1}^{s-1} \mathbb{E} \parallel \theta_{q,n,j-1} - \theta_q \parallel^2 + 8\eta^2(s-1)^2 \mathbb{E} \parallel \nabla F(\theta_q) \parallel^2$$

$$B_5^{(q)} \leq 2w^2 \mathbb{E} \parallel \theta_q \parallel^2$$

$$= 2w^2 \mathbb{E} \parallel \frac{1}{|N_{q-1}^{(i)}|} \sum_{n \in N_{q-1}^{(i)}} \theta_{q-1,n,S} \parallel^2$$

$$\leq \frac{2w^2}{|N_{q-1}^{(i)}|} \sum_{n \in N_{q-1}^{(i)}} \mathbb{E} \parallel \theta_{q-1,n,0} - \eta \sum_{j=0}^{S-1} \nabla f_n(\theta_{q-1,n,j}, \xi_{n,j}) \odot m_{q-1,n} \parallel^2$$

$$\leq 2w^2 \left( \frac{2}{|N_{q-1}^{(i)}|} \sum_{n \in N_{q-1}^{(i)}} \mathbb{E} \parallel \theta_{q-1} \odot m_{q-1,n} \parallel^2 + \frac{2\eta^2}{|N_{q-1}^{(i)}|} \sum_{n \in N_{q-1}^{(i)}} \mathbb{E} \parallel \sum_{j=0}^{S-1} \nabla f_n(\theta_{q-1,n,j}, \xi_{n,j}) \parallel^2 \right)$$

When bounding the term $B_4^{(q)}$, we avoid the bounded gradient assumption used by Zhou et al. (2024) due to the complexity of the model evolution trajectory in practice. Instead, we utilize the existing Lipschitzian gradient and bounded variance assumptions, also proposed in their work, to derive the bound. Additionally, we have made a more refined treatment of the bound of $B_5^{(q)}$, relaxing it to the scaled sum of the accumulation of $B_2^{(q)}$ over rounds and the norm of the initial model. This treatment makes the final result independent of the average model norm throughout training, improving upon the work of Zhou et al. (2024). This improvement played a key role in the subsequent revelation of the impact of complex factors on the local score estimation error.

Next, we further bound $\sum_{q=0}^{Q-1} B_3^{(q)}$ as follows:

$$\sum_{q=0}^{Q-1} B_3^{(q)} \leq \eta^2 S L^2 \cdot \sum_{q=0}^{Q-1} \frac{1}{|N_q^{(i)}|} \sum_{n \in N_q^{(i)}} \sum_{s=1}^{S} \mathbb{E} \parallel \nabla\theta_{q,n,s-1} - \nabla\theta_q \parallel^2$$

$$\leq \frac{\eta^2 S^2 Q}{2}(\sigma_1^2 + \sigma_2^2) + \frac{6\eta^2 S^2 L^2 w^4}{1 - 2w^2} \mathbb{E} \parallel \theta_0 \parallel^2 + \frac{\eta^2 S^2}{2} \sum_{q=0}^{Q-1} \mathbb{E} \parallel \nabla F(\theta_q) \parallel^2 \tag{14}$$

By summing $B_1^{(q)}$ and $B_2^{(q)}$ from $q = 0$ to $Q - 1$, and substituting $B_3^{(q)}$ into both terms, we can then select an appropriate step size $\eta$ to obtain the final convergence result.

## 5.2 PROOF SKETCH OF COROLLARY 2

According to Corollary 1, we can obtain the following result:

$$
\begin{aligned}
&F(\theta_Q) \\
&= \mathcal{O}\Big(F(\theta_0) + (\frac{\sqrt{\Gamma^* S Q} L^2 w^4}{3Q(1 - 2w^2)} + \frac{3\sqrt{\Gamma^* S Q} L^2 w^4}{Q(1 - 2w^2)})\mathbb{E} \parallel \theta_0 \parallel^2 + \frac{3LN\sigma_1^2}{2\Gamma^*} + \frac{5\sqrt{\Gamma^* S Q}}{18}(\sigma_1^2 + \sigma_2^2)\Big)
\end{aligned}
$$
(15)

where $F(\theta_Q) = \frac{1}{N} \sum_{n=1}^{N} \sum_{k=0}^{K-1} \gamma_k \mathbb{E} \parallel s_{\theta_Q}(Y_{n,t_k}, T - t_k) - \nabla \log q(Y_{n,t_k} | X_{n,0}) \parallel^2$ represents the global loss on the trained score estimation model $\theta_Q$.

Therefore, to relax the constant assumption on the local score estimation error (Benton et al., 2024), which is $\sum_{k=0}^{K-1} \gamma_k \mathbb{E} \parallel s_{\theta_Q}(Y_{n,t_k}, T - t_k) - \nabla \log q(Y_{n,t_k}) \parallel^2 \leq \epsilon_{\text{score}}^2$, we must additionally account for two types of errors: **the loss error introduced by denoising score matching, and the discrepancy between the global loss $F(\theta_Q)$ and the local loss $F_n(\theta_Q)$.**

The former is discussed in detail in Appendix B, and we only list the result, which is

$$
\sum_{k=0}^{K-1} \gamma_k \mathbb{E} \parallel s_{\theta_Q}(Y_{n,t_k}, T - t_k) - \nabla \log q(Y_{n,t_k}) \parallel^2
$$

$$
\leq \sum_{k=0}^{K-1} \gamma_k \mathbb{E} \parallel s_{\theta_Q}(Y_{n,t_k}, T - t_k) - \nabla \log q(Y_{n,t_k} | X_{n,0}) \parallel^2 + \sum_{k=0}^{K-1} \gamma_k C = F_n(\theta_Q) + C(T - \delta)
$$

where $C$ is a constant. As for the latter, through the constructor $h(t) = \theta_0 + t(\theta_Q - \theta_0)$, it holds that

$$
F(\theta_Q) - F(\theta_0) = \int_0^1 \nabla F(h(t))^T (\theta_Q - \theta_0) dt
$$
(16)

$$
F_n(\theta_Q) - F_n(\theta_0) = \int_0^1 \nabla F_n(h(t))^T (\theta_Q - \theta_0) dt
$$
(17)

By subtracting the two equations, applying the norm and the bounded variance assumption, we obtain

$$
\parallel F_n(\theta_Q) - F(\theta_Q) \parallel \leq \parallel F_n(\theta_0) - F(\theta_0) \parallel + \sigma_2 \parallel \theta_Q - \theta_0 \parallel
$$
(18)

By utilizing the aforementioned inequalities, we derive Corollary 2, which extends the theoretical results of Benton et al. (2024) on diffusion models in the single-worker paradigm to resource-constrained distributed scenarios.

## 6 CONCLUSION

In this paper, we provide the first generation error bound for distributed diffusion models, without assuming perfect score approximation. This theoretical bound is linear in the data dimension $d$, aligning with state-of-the-art results from the single-worker paradigm. Furthermore, it theoretically demonstrates how distributed training dynamics affect generation performance.

Our work enhances theoretical understanding of distributed diffusion models, it also reveals some interesting phenomena. For example, as discussed in Remark 1, suitable $Q$ helps tighten the bound on $\mathcal{O}(\epsilon^2)$. This depends on the specific scenario, i.e., the target loss function of all workers is the same and the error caused by random sampling is negligible. This also shows that the diffusion model training has low tolerance for errors.

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

# A   NOTATION TABLE

In Table 1, we summarize the main notations in this paper.

Table 1: Notations and Descriptions

| Notations | Descriptions |
|---|---|
| $T$ | The total time of noise scheduling |
| $t$ | The current time of noise scheduling |
| $K$ | The total number of discretized time interval of noise scheduling |
| $t_k$ | The $k$-th discretized time point of noise scheduling, and it holds $0 = t_0 < t_1 < t_2 < \cdots < t_K \leq T$ |
| $X_{n,t}$ | The data of worker-$n$ at time $t$ of noise scheduling, such as image data |
| $Y_{n,t}$ | The data of worker-$n$, which satisfies $Y_{n,t} = X_{n,T-t}$ |
| $q_t, t \in [0, T]$ | The marginals of the forward process |
| $d$ | The dimension of data |
| $(B_t)_{t \in [0,T]}$ | The standard Brownian motion on $\mathbb{R}^d$ |
| $(\tilde{B}_t)_{t \in [0,T]}$ | The standard Brownian motion on $\mathbb{R}^d$ |
| $(B'_t)_{t \in [0,T]}$ | The standard Brownian motion on $\mathbb{R}^d$ |
| $s_\theta(X_t, t)$ | The score approximation which can be parameterized by a neural network with a parameter vector $\theta \in \mathbb{R}^D$ |
| $D$ | The dimension of model parameter $\theta$, $\theta \in \mathbb{R}^D$ |
| $Q$ | The total communication round for training the score approximation $s_\theta$ |
| $q$ | The current communication round for training the score approximation $s_\theta$ |
| $S$ | The number of local steps during two communication rounds |
| $N$ | The total number of workers |
| $N_q^{(i)}$ | The set of workers for which the value of coordinate-$i$ in the mask is non-zero, and $N_q^{(i)} = \{n : m_{q,n}^i = 1\}$ |
| $\Gamma^*$ | The minimum occurrences of any dimension parameter in the local model, and $\Gamma^* = \min_{q,i} |N_q^{(i)}| \geq 1$ |
| $f_n(\theta_{q,n,s}, \xi_{n,s})$ | The loss of worker-$n$ on data sample $\xi_{n,s}$ in the step $s$ of round $q$ |
| $F_n(\theta)$ | The loss function of worker-$n$, and $F_n(\theta) = \mathbb{E}_{\xi_n \sim \mathcal{D}_n}[f_n(\theta, \xi_n)]$ |
| $m_{q,n}$ | The local mask of worker-$n$ generated by mask policy, and $m_{q,n} \in \{0, 1\}^D$ |
| $\eta$ | The step size for training the score approximation $s_\theta$ |

# B   EQUIVALENT OBJECTIVE WITH DENOISING SCORE MATCHING

First, we considered the following loss function:

$$\frac{1}{N} \sum_{n=1}^N \sum_{k=0}^{K-1} \gamma_k \mathbb{E} \parallel s_\theta(Y_{n,t_k}, T - t_k) - \nabla \log q(Y_{n,t_k}) \parallel^2 \tag{19}$$

where $\sum_{k=0}^{K-1} \gamma_k \mathbb{E} \parallel s_\theta(Y_{n,t_k}, T - t_k) - \nabla \log q(Y_{n,t_k}) \parallel^2$ can be considered as the time-discretized version of the loss function (4). Since the score function $\nabla \log q_{n,t}(\cdot)$, we alternatively consider a denoising score matching objective, which is derived following:

$$\mathbb{E} \parallel s_\theta(X_{n,t}) - \nabla \log q(X_{n,t}) \parallel^2$$
$$= \mathbb{E} \parallel s_\theta(X_{n,t}) \parallel^2 + \mathbb{E} \parallel \nabla \log q(X_{n,t}) \parallel^2 - 2\mathbb{E} \langle s_\theta(X_{n,t}), \nabla \log q(X_{n,t}) \rangle$$
$$= \mathbb{E} \parallel s_\theta(X_{n,t}) \parallel^2 + \mathbb{E} \parallel \nabla \log q(X_{n,t}) \parallel^2 - 2\mathbb{E}_{q_{n,0}} \mathbb{E}_{q_{n,t|0}} \langle s_\theta(X_{n,t}), \nabla \log q_{n,t|0}(X_{n,t}|X_{n,0}) \rangle$$
$$= \mathbb{E} \parallel s_\theta(X_{n,t}) \parallel^2 + \mathbb{E} \parallel \nabla \log q(X_{n,t}) \parallel^2 + 2\mathbb{E}_{q_{n,0}} \mathbb{E}_{q_{n,t|0}} \langle s_\theta(X_{n,t}), \frac{X_{n,t} - e^{-t} X_{n,0}}{1 - e^{-2t}} \rangle$$
$$= \mathbb{E} \parallel s_\theta(X_{n,t}) + \frac{X_{n,t} - e^{-t} X_{n,0}}{1 - e^{-2t}} \parallel^2 + \mathbb{E} \parallel \nabla \log q(X_{n,t}) \parallel^2 - \frac{d}{1 - e^{-2t}}$$

$$=\mathbb{E} \parallel s_\theta(X_{n,t},t) + \frac{X_{n,t} - e^{-t}X_{n,0}}{1 - e^{-2t}} \parallel^2 + C_t$$

$$=\mathbb{E} \parallel s_\theta(X_{n,t},t) - \nabla \log q_{n,t|0}(X_{n,t}|X_{n,0}) \parallel^2 + C_t \tag{20}$$

where $C_t$ is a constant independent of $\theta$. Let $C = \max_t C_t$, then it holds that

$$\frac{1}{N} \sum_{n=1}^{N} \sum_{k=0}^{K-1} \gamma_k \mathbb{E} \parallel s_\theta(Y_{n,t_k}, T - t_k) - \nabla \log q(Y_{n,t_k}) \parallel^2$$

$$\leq \frac{1}{N} \sum_{n=1}^{N} \sum_{k=0}^{K-1} \gamma_k \mathbb{E} \parallel s_\theta(Y_{n,t_k}, T - t_k) - \nabla \log q(Y_{n,t_k}|X_{n,0}) \parallel^2 + \frac{1}{N} \sum_{n=1}^{N} \sum_{k=0}^{K-1} \gamma_k C$$

$$=\frac{1}{N} \sum_{n=1}^{N} \left( F_n(\theta) + C(T - \delta) \right) \tag{21}$$

Therefore, as measures of learning loss, Equations (9) and (19) are equivalent because the only difference between them is a constant.

## C    SOLUTION TO EQUATION (10)

Consider the Equation (10):

$$d\widetilde{Y}_{n,t} = \{\widetilde{Y}_{n,t} + 2s_{\theta_Q}(\widetilde{Y}_{n,t_k}, T - t_k)\}dt + \sqrt{2}d\widetilde{B}_{n,t}$$

And we multiply both sides of the Equation (10) by $e^{-t}$ to get

$$d(e^{-t}\widetilde{Y}_{n,t}) = -2s_{\theta_Q}(\widetilde{Y}_{n,t_k}, T - t_k)\}d(e^{-t}) + \sqrt{2}e^{-t}d\widetilde{B}_{n,t} \tag{22}$$

For each time interval $[t_k, t_{k+1}]$, we perform an integration operation to derive the following result:

$$e^{-t_{k+1}}\widetilde{Y}_{n,t_{k+1}} = e^{-t_k}\widetilde{Y}_{n,t_k} + 2s_{\theta_Q}(\widetilde{Y}_{n,t_k}, T - t_k)\}(e^{-t_k} - e^{-t_{k+1}}) + \sqrt{2} \int_{t_k}^{t_{k+1}} e^{-t}d\widetilde{B}_{n,t} \tag{23}$$

And then the following Equation (24) can be derived by multiplying both sides of the Equation (23) by $e^{t_{k+1}}$:

$$\widetilde{Y}_{n,t_{k+1}} = e^{\gamma_k}\widetilde{Y}_{n,t_k} + 2(e^{\gamma_k} - 1)s_{\theta_Q}(\widetilde{Y}_{n,t_k}, T - t_k) + \sqrt{e^{2\gamma_k} - 1}\epsilon_{n,k} \tag{24}$$

where $\gamma_k = t_{k+1} - t_k$ and $\epsilon_{n,k} \sim \mathcal{N}(\mathbf{0}, \boldsymbol{I}_d)$. And Equation (24) is exactly the solution to Equation (10).

## D    PROOF OF THEOREM 1

Building on Assumption 1, we can straightforwardly deduce that the function $F(\cdot)$ is also $L$-smooth, satisfying the following inequality:

$$\mathbb{E}[F(\theta_{q+1})] - \mathbb{E}[F(\theta_q)] \leq \underbrace{\mathbb{E}\langle \nabla F(\theta_q), \theta_{q+1} - \theta_q\rangle}_{B_1^{(q)}} + \underbrace{\frac{L}{2}\mathbb{E} \parallel \theta_{q+1} - \theta_q \parallel^2}_{B_2^{(q)}} \tag{25}$$

Now, we consider the situation where $\Gamma^* \geq 1$, and we first discuss the bound of $B_1^{(q)}$:

$$B_1^{(q)} = \mathbb{E}\langle \nabla F(\theta_q), \theta_{q+1} - \theta_q\rangle$$

$$= \sum_{i=1}^{D} \mathbb{E}\langle \nabla F^{(i)}(\theta_q), \theta_{q+1}^{(i)} - \theta_q^{(i)}\rangle$$

$$= \sum_{i=1}^{D} \mathbb{E} \langle \nabla F^{(i)}(\theta_q), -\frac{1}{|N_q^{(i)}|} \sum_{n \in N_q^{(i)}} \sum_{s=1}^{S} \eta \nabla f_n^{(i)}(\theta_{q,n,s-1}, \xi_{n,s-1}) \rangle$$

$$= \sum_{i=1}^{D} \mathbb{E} \langle \nabla F^{(i)}(\theta_q), -\frac{1}{|N_q^{(i)}|} \sum_{n \in N_q^{(i)}} \sum_{s=1}^{S} \eta \nabla F_n^{(i)}(\theta_{q,n,s-1}) \rangle$$

$$= - \sum_{i=1}^{D} \mathbb{E} \langle \nabla F^{(i)}(\theta_q), \frac{1}{|N_q^{(i)}|} \sum_{n \in N_q^{(i)}} \sum_{s=1}^{S} \eta [\nabla F_n^{(i)}(\theta_{q,n,s-1}) - \nabla F_n^{(i)}(\theta_q)] \rangle$$

$$+ \sum_{i=1}^{D} \mathbb{E} \langle \nabla F^{(i)}(\theta_q), -\eta S \nabla F^{(i)}(\theta_q) \rangle$$

$$\leq \frac{\eta}{2S} \sum_{i=1}^{D} \mathbb{E} \| \frac{1}{|N_q^{(i)}|} \sum_{n \in N_q^{(i)}} \sum_{s=1}^{S} [\nabla F_n^{(i)}(\theta_{q,n,s-1}) - \nabla F_n^{(i)}(\theta_q)] \|^2$$

$$- S\eta \mathbb{E} \| \nabla F(\theta_q) \|^2 + \frac{S\eta}{2} \sum_{i=1}^{D} \mathbb{E} \| \nabla F^{(i)}(\theta_q) \|^2$$

$$= - \frac{S\eta}{2} \mathbb{E} \| \nabla F(\theta_q) \|^2 + \frac{\eta}{2S} \sum_{i=1}^{D} \mathbb{E} \| \frac{1}{|N_q^{(i)}|} \sum_{n \in N_q^{(i)}} \sum_{s=1}^{S} [\nabla F_n^{(i)}(\theta_{q,n,s-1}) - \nabla F_n^{(i)}(\theta_q)] \|^2$$

$$= - \frac{S\eta}{2} \mathbb{E} \| \nabla F(\theta_q) \|^2 + \frac{1}{2S\eta} B_3^{(q)} \tag{26}$$

where

$$B_3^{(q)} = \sum_{i=1}^{D} \mathbb{E} \| \frac{\eta}{|N_q^{(i)}|} \sum_{n \in N_q^{(i)}} \sum_{s=1}^{S} [\nabla F_n^{(i)}(\theta_{q,n,s-1}) - \nabla F_n^{(i)}(\theta_q)] \|^2$$

And we next consider how to bound $B_2^{(q)}$:

$$B_2^{(q)} = \frac{L}{2} \sum_{i=1}^{D} \mathbb{E} \| \theta_{q+1}^{(i)} - \theta_q^{(i)} \|^2$$

$$= \frac{L}{2} \sum_{i=1}^{D} \mathbb{E} \| \frac{1}{|N_q^{(i)}|} \sum_{n \in N_q^{(i)}} \theta_{q,n,S}^{(i)} - \theta_q^{(i)} \|^2$$

$$= \frac{L}{2} \sum_{i=1}^{D} \mathbb{E} \| \frac{1}{|N_q^{(i)}|} \sum_{n \in N_q^{(i)}} \left( \theta_{q,n,S-1}^{(i)} - \eta \nabla f_n^{(i)}(\theta_{q,n,S-1}, \xi_{n,S-1}) \cdot m_{q,n}^{(i)} \right) - \theta_q^{(i)} \|^2$$

$$= \frac{L}{2} \sum_{i=1}^{D} \mathbb{E} \| -\frac{1}{|N_q^{(i)}|} \sum_{n \in N_q^{(i)}} \sum_{s=1}^{S} \eta \nabla f_n^{(i)}(\theta_{q,n,s-1}, \xi_{n,s-1}) + \frac{1}{|N_q^{(i)}|} \sum_{n \in N_q^{(i)}} \theta_{q,n,0}^{(i)} - \theta_q^{(i)} \|^2$$

$$= \frac{L}{2} \sum_{i=1}^{D} \mathbb{E} \| -\frac{1}{|N_q^{(i)}|} \sum_{n \in N_q^{(i)}} \sum_{s=1}^{S} \eta \nabla f_n^{(i)}(\theta_{q,n,s-1}, \xi_{n,s-1}) \|^2$$

$$= \frac{L}{2} \sum_{i=1}^{D} \mathbb{E} \| -\frac{1}{|N_q^{(i)}|} \sum_{n \in N_q^{(i)}} \sum_{s=1}^{S} \eta \left( \nabla f_n^{(i)}(\theta_{q,n,s-1}, \xi_{n,s-1}) - \nabla F_n^{(i)}(\theta_{q,n,s-1}) + \nabla F_n^{(i)}(\theta_{q,n,s-1}) \right.$$

$$\left. - \nabla F_n^{(i)}(\theta_q) + \nabla F_n^{(i)}(\theta_q) \|^2$$

$$\leq \frac{3L\eta^2}{2} \sum_{i=1}^{D} \mathbb{E} \| \frac{1}{|N_q^{(i)}|} \sum_{n \in N_q^{(i)}} \sum_{s=1}^{S} [\nabla f_n^{(i)}(\theta_{q,n,s-1}, \xi_{n,s-1}) - \nabla F_n^{(i)}(\theta_{q,n,s-1})] \|^2$$

$$+\frac{3L}{2}\sum_{i=1}^{D}\mathbb{E}\parallel\frac{\eta}{|N_q^{(i)}|}\sum_{n\in N_q^{(i)}}\sum_{s=1}^{S}\left(\nabla F_n^{(i)}(\theta_{q,n,s-1})-\nabla F_n^{(i)}(\theta_q)\right)\parallel^2$$

$$+\frac{3L}{2}\sum_{i=1}^{D}\mathbb{E}\parallel\frac{1}{|N_q^{(i)}|}\sum_{n\in N_q^{(i)}}\sum_{s=1}^{S}\eta\nabla F_n^{(i)}(\theta_q)\parallel^2$$

$$\leq\frac{3NLS\eta^2\sigma_1^2}{2(\Gamma^*)^2}+\frac{3LS^2\eta^2}{2}\mathbb{E}\parallel\nabla F(\theta_q)\parallel^2+\frac{3L}{2}B_3^{(q)}\tag{27}$$

Therefore, discussing the bound of $B_3^{(q)}$ will help us explore $B_1^{(q)}$ and $B_2^{(q)}$:

$$B_3^{(q)}=\sum_{i=1}^{D}\mathbb{E}\parallel\frac{\eta}{|N_q^{(i)}|}\sum_{n\in N_q^{(i)}}\sum_{s=1}^{S}[\nabla F_n^{(i)}(\theta_{q,n,s-1})-\nabla F_n^{(i)}(\theta_q)]\parallel^2$$

$$\leq\sum_{i=1}^{D}\frac{S\eta^2}{|N_q^{(i)}|}\sum_{n\in N_q^{(i)}}\sum_{s=1}^{S}\mathbb{E}\parallel\nabla F_n^{(i)}(\theta_{q,n,s-1})-\nabla F_n^{(i)}(\theta_q)\parallel^2$$

$$\leq\eta^2SL^2\cdot\frac{1}{|N_q^{(i)}|}\sum_{n\in N_q^{(i)}}\sum_{s=1}^{S}\mathbb{E}\parallel\nabla\theta_{q,n,s-1}-\nabla\theta_q\parallel^2\tag{28}$$

And it holds that

$$\mathbb{E}\parallel\theta_{q,n,s-1}-\theta_q\parallel^2\leq2\mathbb{E}\parallel\theta_{q,n,s-1}-\theta_{q,n,0}\parallel^2+2\mathbb{E}\parallel\theta_{q,n,0}-\theta_q\parallel^2$$

$$=2\mathbb{E}\parallel\theta_{q,n,s-1}-\theta_{q,n,0}\parallel^2+2\mathbb{E}\parallel\theta_q\odot m_{n,q}-\theta_q\parallel^2$$

$$=\underbrace{2\mathbb{E}\parallel\theta_{q,n,s-1}-\theta_{q,n,0}\parallel^2}_{B_4^{(q)}}+\underbrace{2w^2\mathbb{E}\parallel\theta_q\parallel^2}_{B_5^{(q)}}$$

We bound $B_4^{(q)}$ and $B_5^{(q)}$ separately:

$$B_4^{(q)}=2\mathbb{E}\parallel-\eta\sum_{j=0}^{s-2}\nabla f_n(\theta_{q,n,j},\xi_{n,j})\odot m_{q,n}\parallel^2$$

$$\leq2\eta^2(s-1)\sum_{j=1}^{s-1}\mathbb{E}\parallel\nabla f_n(\theta_{q,n,j-1},\xi_{n,j-1})-\nabla F_n(\theta_{q,n,j-1})+\nabla F_n(\theta_{q,n,j-1})-\nabla F_n(\theta_q)$$

$$+\nabla F_n(\theta_q)-\nabla F(\theta_q)+\nabla F(\theta_q)\parallel^2$$

$$\leq8\eta^2(s-1)^2(\sigma_1^2+\sigma_2^2)+8\eta^2L^2(s-1)\sum_{j=1}^{s-1}\mathbb{E}\parallel\theta_{q,n,j-1}-\theta_q\parallel^2+8\eta^2(s-1)^2\mathbb{E}\parallel\nabla F(\theta_q)\parallel^2$$

$$\color{red}{\mathbb{E}\parallel\theta_q\parallel^2}$$

$$\color{red}{=\mathbb{E}\parallel\frac{1}{|N_{q-1}^{(i)}|}\sum_{n\in N_{q-1}^{(i)}}\theta_{q-1,n,S}\parallel^2}$$

$$\color{red}{\leq\frac{1}{|N_{q-1}^{(i)}|}\sum_{n\in N_{q-1}^{(i)}}\mathbb{E}\parallel\theta_{q-1,n,0}-\eta\sum_{j=0}^{S-1}\nabla f_n(\theta_{q-1,n,j},\xi_{n,j})\odot m_{q-1,n}\parallel^2}$$

$$\color{red}{\leq\frac{2}{|N_{q-1}^{(i)}|}\sum_{n\in N_{q-1}^{(i)}}\mathbb{E}\parallel\theta_{q-1}\odot m_{q-1,n}\parallel^2+\frac{2\eta^2}{|N_{q-1}^{(i)}|}\sum_{n\in N_{q-1}^{(i)}}\mathbb{E}\parallel\sum_{j=0}^{S-1}\nabla f_n(\theta_{q-1,n,j},\xi_{n,j})\parallel^2}$$

$$\leq 2w^2 \mathbb{E} \parallel \theta_{q-1} \parallel^2 + \frac{2\eta^2 S}{|N_{q-1}^{(i)}|} \sum_{n \in N_{q-1}^{(i)}} \sum_{j=0}^{S-1} \mathbb{E} \parallel \nabla f_n(\theta_{q-1,n,j}, \xi_{n,j}) \parallel^2$$

$$\leq 2w^2 \mathbb{E} \parallel \theta_{q-1} \parallel^2 + \frac{2\eta^2 S}{|N_{q-1}^{(i)}|} \sum_{n \in N_{q-1}^{(i)}} \sum_{j=0}^{S-1} \mathbb{E} \parallel \nabla f_n(\theta_{q-1,n,j}, \xi_{n,j}) - \nabla F_n(\theta_{q-1,n,j}) +$$

$$\nabla F_n(\theta_{q-1,n,j}) - \nabla F_n(\theta_{q-1}) + \nabla F_n(\theta_{q-1}) - \nabla F(\theta_{q-1}) + \nabla F(\theta_{q-1}) \parallel^2$$

$$\leq 2w^2 \mathbb{E} \parallel \theta_{q-1} \parallel^2 + 8\eta^2 S^2 (\sigma_1^2 + \sigma_2^2) + \frac{8\eta^2 SL^2}{|N_{q-1}^{(i)}|} \sum_{n \in N_{q-1}^{(i)}} \sum_{j=0}^{S-1} \mathbb{E} \parallel \theta_{q-1,n,j} - \theta_{q-1} \parallel^2$$

$$+ 8\eta^2 S^2 \mathbb{E} \parallel \nabla F(\theta_{q-1}) \parallel^2$$

Summing from $q = 1$ to $Q$ for $\mathbb{E} \parallel \theta_q \parallel^2$ yields

$$\sum_{q=1}^{Q} \mathbb{E} \parallel \theta_q \parallel^2$$

$$\leq 2w^2 \sum_{q=1}^{Q} \mathbb{E} \parallel \theta_{q-1} \parallel^2 + 8\eta^2 S^2 \sum_{q=1}^{Q} (\sigma_1^2 + \sigma_2^2) + \sum_{q=1}^{Q} \frac{8\eta^2 SL^2}{|N_{q-1}^{(i)}|} \sum_{n \in N_{q-1}^{(i)}} \sum_{j=0}^{S-1} \mathbb{E} \parallel \theta_{q-1,n,j} - \theta_{q-1} \parallel^2$$

$$+ 8\eta^2 S^2 \sum_{q=1}^{Q} \mathbb{E} \parallel \nabla F(\theta_{q-1}) \parallel^2$$

Therefore, we have

$$(1 - 2w^2) \sum_{q=1}^{Q} \mathbb{E} \parallel \theta_q \parallel^2$$

$$\leq 2w^2 \mathbb{E} \parallel \theta_0 \parallel^2 + 8\eta^2 S^2 \sum_{q=1}^{Q} (\sigma_1^2 + \sigma_2^2) + \sum_{q=0}^{Q-1} \frac{8\eta^2 SL^2}{|N_q^{(i)}|} \sum_{n \in N_q^{(i)}} \sum_{j=0}^{S-1} \mathbb{E} \parallel \theta_{q,n,j} - \theta_q \parallel^2$$

$$+ 8\eta^2 S^2 \sum_{q=0}^{Q-1} \mathbb{E} \parallel \nabla F(\theta_q) \parallel^2 \tag{29}$$

Summing from $s = 1$ to $S$ for Eq. (29) yields

$$(1 - 2w^2) \sum_{q=1}^{Q} \sum_{s=1}^{S} \mathbb{E} \parallel \theta_q \parallel^2$$

$$\leq 2w^2 S \mathbb{E} \parallel \theta_0 \parallel^2 + 8\eta^2 S^3 \sum_{q=1}^{Q} (\sigma_1^2 + \sigma_2^2) + \sum_{q=0}^{Q-1} \frac{8\eta^2 S^2 L^2}{|N_q^{(i)}|} \sum_{n \in N_q^{(i)}} \sum_{s=0}^{S-1} \mathbb{E} \parallel \theta_{q,n,s} - \theta_q \parallel^2$$

$$+ 8\eta^2 S^3 \sum_{q=0}^{Q-1} \mathbb{E} \parallel \nabla F(\theta_q) \parallel^2 \tag{30}$$

Next summing from $s = 1$ to $S$ and $q = 1$ to $Q$ for $\mathbb{E} \parallel \theta_{q,n,s-1} - \theta_q \parallel^2$, then substituting Eq.(30) into it yields

$$\sum_{q=1}^{Q} \sum_{s=1}^{S} \mathbb{E} \parallel \theta_{q,n,s-1} - \theta_q \parallel^2$$

$$\leq 2\mathbb{E}\parallel \theta_{q,n,s-1} - \theta_{q,n,0} \parallel^2 + 2\mathbb{E}\parallel \theta_{q,n,0} - \theta_q \parallel^2$$

$$\leq \sum_{q=1}^{Q}\sum_{s=1}^{S} B_4^{(q)} + \sum_{q=1}^{Q}\sum_{s=1}^{S} B_5^{(q)}$$

$$= \sum_{q=1}^{Q}\sum_{s=1}^{S} B_4^{(q)} + 2w^2\sum_{q=1}^{Q}\sum_{s=1}^{S}\mathbb{E}\parallel \theta_q \parallel^2$$

$$\leq 8\eta^2 S^3 \sum_{q=1}^{Q}(\sigma_1^2 + \sigma_2^2) + 8\eta^2 L^2 S^2 \sum_{q=1}^{Q}\sum_{s=1}^{S}\mathbb{E}\parallel \theta_{q,n,s-1} - \theta_q \parallel^2 + 8\eta^2 S^3 \sum_{q=1}^{Q}\mathbb{E}\parallel \nabla F(\theta_q) \parallel^2$$

$$+ \frac{2w^2}{1-2w^2}(2w^2 S\mathbb{E}\parallel \theta_0 \parallel^2 + 8\eta^2 S^3\sum_{q=1}^{Q}(\sigma_1^2 + \sigma_2^2) + \sum_{q=0}^{Q-1}\frac{8\eta^2 S^2 L^2}{|N_q^{(i)}|}\sum_{n\in N_q^{(i)}}\sum_{s=0}^{S-1}\mathbb{E}\parallel \theta_{q,n,s} - \theta_q \parallel^2$$

$$+ 8\eta^2 S^3 \sum_{q=0}^{Q-1}\mathbb{E}\parallel \nabla F(\theta_q) \parallel^2) \tag{31}$$

Summing all $n \in N_q^{(i)}$ for Eq. (31) yields

$$\sum_{q=1}^{Q}\sum_{n\in N_q^{(i)}}\sum_{s=1}^{S}\mathbb{E}\parallel \theta_{q,n,s-1} - \theta_q \parallel^2$$

$$\leq 8\eta^2 S^3 \sum_{q=1}^{Q}|N_q^{(i)}|(\sigma_1^2 + \sigma_2^2) + 8\eta^2 L^2 S^2 \sum_{q=1}^{Q}\sum_{s=1}^{S}\sum_{n\in N_q^{(i)}}\mathbb{E}\parallel \theta_{q,n,s-1} - \theta_q \parallel^2 + 8\eta^2 S^3 \sum_{q=1}^{Q}|N_q^{(i)}|\mathbb{E}\parallel \nabla F(\theta_q) \parallel^2$$

$$+ \frac{4w^4 S}{1-2w^2}|N_q^{(i)}|\mathbb{E}\parallel \theta_0 \parallel^2 + \frac{16\eta^2 S^3 w^2 \sum_{q=1}^{Q}|N_q^{(i)}|(\sigma_1^2 + \sigma_2^2)}{1-2w^2} + \frac{16}{\eta^2 S^3 w^2}\sum_{q=0}^{Q-1}|N_q^{(i)}|\mathbb{E}\parallel \nabla F(\theta_q) \parallel^2$$

$$+ \frac{16\eta^2 S^2 L^2 w^2}{1-2w^2}\sum_{q=0}^{Q-1}\sum_{n\in N_q^{(i)}}\sum_{s=1}^{S}\mathbb{E}\parallel \theta_{q,n,s-1} - \theta_q \parallel^2 \tag{32}$$

Let $H_0 = 1 - 8\eta^2 L^2 S^2 - \frac{16\eta^2 S^2 L^2 w^2}{1-2w^2}$, then Eq. (32) can be rewritten as

$$H_0 \sum_{q=1}^{Q}\sum_{n\in N_q^{(i)}}\sum_{s=1}^{S}\mathbb{E}\parallel \theta_{q,n,s-1} - \theta_q \parallel^2$$

$$\leq (8\eta^2 S^3 + \frac{16\eta^2 S^3 w^2}{1-2w^2})\sum_{q=1}^{Q}|N_q^{(i)}|(\sigma_1^2 + \sigma_2^2) + \frac{4w^4 S}{1-2w^2}|N_q^{(i)}|\mathbb{E}\parallel \theta_0 \parallel^2$$

$$+ (8\eta^2 S^3 + \frac{16\eta^2 S^3 w^2}{1-2w^2})\sum_{q=0}^{Q-1}|N_q^{(i)}|\mathbb{E}\parallel \nabla F(\theta_q) \parallel^2$$

Let $H_0 = 1 - 8\eta^2 L^2 S^2 - \frac{16\eta^2 S^2 L^2 w^2}{1-2w^2} \geq \frac{2}{3} \Leftarrow \eta^2 \leq \frac{1}{3(8L^2 S^2 + \frac{16 S^2 L^2 w^2}{1-2w^2})}$, then it holds

$$\frac{1}{H_0} \leq \frac{3}{2}$$

$$8\eta^2 L^2 S^2 + \frac{16\eta^2 S^2 L^2 w^2}{1-2w^2} \leq \frac{1}{3}$$

Then we can further derive

$$\sum_{q=1}^{Q}\sum_{n\in N_q^{(i)}}\sum_{s=1}^{S}\mathbb{E}\parallel \theta_{q,n,s-1} - \theta_q \parallel^2$$

$$\leq \frac{S}{2L^2} \sum_{q=1}^{Q} |N_q^{(i)}|(\sigma_1^2 + \sigma_2^2) + \frac{6w^4 S}{1 - 2w^2} |N_q^{(i)}| \mathbb{E} \parallel \theta_0 \parallel^2 + \frac{S}{2L^2} \sum_{q=0}^{Q-1} |N_q^{(i)}| \mathbb{E} \parallel \nabla F(\theta_q) \parallel^2 \quad (33)$$

According to Eq. (28)

$$\sum_{q=0}^{Q-1} B_3^{(q)} \leq \eta^2 SL^2 \cdot \sum_{q=0}^{Q-1} \frac{1}{|N_q^{(i)}|} \sum_{n \in N_q^{(i)}} \sum_{s=1}^{S} \mathbb{E} \parallel \nabla \theta_{q,n,s-1} - \nabla \theta_q \parallel^2$$

Substitute Eq. (33) into the above inequality, and we have

$$\sum_{q=0}^{Q-1} B_3^{(q)} \leq \eta^2 SL^2 \cdot \sum_{q=0}^{Q-1} \frac{1}{|N_q^{(i)}|} \sum_{n \in N_q^{(i)}} \sum_{s=1}^{S} \mathbb{E} \parallel \nabla \theta_{q,n,s-1} - \nabla \theta_q \parallel^2$$

$$\leq \frac{\eta^2 S^2 Q}{2}(\sigma_1^2 + \sigma_2^2) + \frac{6\eta^2 S^2 L^2 w^4}{1 - 2w^2} \mathbb{E} \parallel \theta_0 \parallel^2 + \frac{\eta^2 S^2}{2} \sum_{q=0}^{Q-1} \mathbb{E} \parallel \nabla F(\theta_q) \parallel^2 \quad (34)$$

Then it holds that for Eq. (27)

$$\frac{L}{2} \sum_{q=0}^{Q-1} \mathbb{E} \parallel \theta_{q+1} - \theta_q \parallel^2$$

$$= \sum_{q=0}^{Q-1} B_2^{(q)}$$

$$\leq \frac{3\eta^2 SLQN\sigma_1^2}{2(\Gamma^*)^2} + \frac{3\eta^2 S^2 L}{2} \sum_{q=0}^{Q-1} \mathbb{E} \parallel \nabla F(\theta_q) \parallel^2 + \frac{3L}{2} \sum_{q=0}^{Q-1} B_3^{(q)}$$

$$\leq \frac{3\eta^2 SLQN\sigma_1^2}{2(\Gamma^*)^2} + \frac{3\eta^2 S^2 L}{2} \sum_{q=0}^{Q-1} \mathbb{E} \parallel \nabla F(\theta_q) \parallel^2 + \frac{3\eta^2 S^2 LQ}{4}(\sigma_1^2 + \sigma_2^2) + \frac{9\eta^2 S^2 L^3 w^4}{1 - 2w^2} \mathbb{E} \parallel \theta_0 \parallel^2$$

$$+ \frac{3\eta^2 S^2 L}{4} \sum_{q=0}^{Q-1} \mathbb{E} \parallel \nabla F(\theta_q) \parallel^2$$

$$\leq \frac{3\eta^2 SLQN\sigma_1^2}{2(\Gamma^*)^2} + \frac{3\eta^2 S^2 LQ}{4}(\sigma_1^2 + \sigma_2^2) + \frac{9\eta^2 S^2 L^3 w^4}{1 - 2w^2} \mathbb{E} \parallel \theta_0 \parallel^2 + \frac{9\eta^2 S^2 L}{4} \sum_{q=0}^{Q-1} \mathbb{E} \parallel \nabla F(\theta_q) \parallel^2$$

$$(35)$$

And it holds for Eq. (26)

$$\sum_{q=0}^{Q-1} \mathbb{E}\langle \nabla F(\theta_q), \theta_{q+1} - \theta_q \rangle$$

$$= \sum_{q=0}^{Q-1} B_1^{(q)}$$

$$\leq -\frac{S\eta}{2} \sum_{q=0}^{Q-1} \mathbb{E} \parallel \nabla F(\theta_q) \parallel^2 + \frac{1}{2S\eta} \sum_{q=0}^{Q-1} B_3^{(q)}$$

$$\leq -\frac{\eta S}{2} \sum_{q=0}^{Q-1} \mathbb{E} \parallel \nabla F(\theta_q) \parallel^2 + \frac{\eta SQ}{4}(\sigma_1^2 + \sigma_2^2) + \frac{3\eta SL^2 w^4}{1 - 2w^2} \mathbb{E} \parallel \theta_0 \parallel^2 + \frac{\eta S}{4} \sum_{q=0}^{Q-1} \mathbb{E} \parallel \nabla F(\theta_q) \parallel^2$$

$$(36)$$

Then summing from $q = 0$ to $Q - 1$ for Eq. (25) and substituting Eq. (35)-(36) yields

$$F(\theta_Q) - F(\theta_0)$$

$$\leq \sum_{q=0}^{Q-1} \mathbb{E}\langle \nabla F(\theta_q), \theta_{q+1} - \theta_q \rangle + \frac{L}{2} \sum_{q=0}^{Q-1} \mathbb{E} \parallel \theta_{q+1} - \theta_q \parallel^2$$

$$\leq (-\frac{\eta S}{4} + \frac{9\eta^2 S^2 L}{4}) \sum_{q=0}^{Q-1} \mathbb{E} \parallel \nabla F(\theta_q) \parallel^2 + \frac{3\eta^2 SLQN\sigma_1^2}{2(\Gamma^*)^2} + (\frac{3\eta^2 S^2 LQ}{4} + \frac{\eta SQ}{4})(\sigma_1^2 + \sigma_2^2)$$

$$+ (\frac{9\eta^2 S^2 L^3 w^4}{1 - 2w^2} + \frac{3\eta SL^2 w^4}{1 - 2w^2})\mathbb{E} \parallel \theta_0 \parallel^2$$

Let $H_1 = -\frac{\eta S}{4} + \frac{9\eta^2 S^2 L}{4} \leq -\frac{\eta S}{6} \Leftarrow \eta \leq \frac{1}{27SL}$, and multiply both sides of the inequality sign in the above inequality by $\frac{6}{\eta SQ}$ and rearrange the terms around to get

$$\frac{1}{Q} \sum_{q=0}^{Q-1} \mathbb{E} \parallel \nabla F(\theta_q) \parallel^2$$

$$\leq \frac{6(F(\theta_0) - F(\theta_Q))}{\eta SQ} + (\frac{54\eta SL^3 w^4}{Q(1 - 2w^2)} + \frac{18L^2 w^4}{Q(1 - 2w^2)})\mathbb{E} \parallel \theta_0 \parallel^2 + \frac{9\eta LN\sigma_1^2}{(\Gamma^*)^2} +$$

$$(\frac{9\eta SL}{2} + \frac{3}{2})(\sigma_1^2 + \sigma_2^2)$$

where $w \in [0, \frac{\sqrt{2}}{2})$ and $\eta \leq \min\{\frac{1}{27SL}, \sqrt{\frac{1}{3(8L^2 S^2 + \frac{16S^2 L^2 w^2}{1 - 2w^2})}}, 1\}$. This completes the proof of Theorem 1.

## E PROOF OF COROLLARY 1

If $\eta = \sqrt{\frac{\Gamma^*}{SQ}}$, it must satisfy the following inequalities:

$$\sqrt{\frac{\Gamma^*}{SQ}} \leq \frac{1}{27SL} \Rightarrow Q \geq 729\Gamma^* SL^2$$

$$\sqrt{\frac{\Gamma^*}{SQ}} \leq \sqrt{\frac{1}{3(8L^2 S^2 + \frac{16S^2 L^2 w^2}{1 - 2w^2})}} \Rightarrow Q \geq 3\Gamma^*(8SL^2 + \frac{16SL^2 w^2}{1 - 2w^2})$$

And if we further make $Q \geq \Gamma^* S$, we have $\sqrt{\Gamma^* S} \leq \sqrt{Q}$.

Using the relationship $\frac{54\eta SL^3 w^4}{Q(1-2w^2)} = \frac{54SL^3 w^4}{Q(1-2w^2)} \cdot \frac{1}{27SL} = \frac{2L^2 w^4}{Q(1-2w^2)}$ and $\frac{9\eta SL}{2} = \frac{9SL}{2} \cdot \frac{1}{27SL} = \frac{1}{6}$, we have

$$\frac{1}{Q} \sum_{q=0}^{Q-1} \mathbb{E} \parallel \nabla F(\theta_q) \parallel^2$$

$$\leq \frac{6(F(\theta_0) - F(\theta_Q))}{\sqrt{\Gamma^* SQ}} + (\frac{2L^2 w^4}{Q(1 - 2w^2)} + \frac{18L^2 w^4}{Q(1 - 2w^2)})\mathbb{E} \parallel \theta_0 \parallel^2 + \frac{9LN\sigma_1^2}{\Gamma^* \sqrt{\Gamma^* SQ}} + \frac{5}{3}(\sigma_1^2 + \sigma_2^2)$$

where $\frac{1}{\sqrt{\Gamma^* SQ}}$ dominates the convergence rate.

## F PROOF OF COROLLARY 2

According to Corollary 1, we can obtain the following result:

$$F(\theta_Q)$$

$$=\mathcal{O}\Big(F(\theta_0) + \big(\frac{\sqrt{\Gamma^* S Q} L^2 w^4}{3Q(1-2w^2)} + \frac{3\sqrt{\Gamma^* S Q} L^2 w^4}{Q(1-2w^2)}\big)\mathbb{E}\parallel \theta_0 \parallel^2 + \frac{3LN\sigma_1^2}{2\Gamma^*} + \frac{5\sqrt{\Gamma^* S Q}}{18}(\sigma_1^2 + \sigma_2^2)\Big) \tag{37}$$

Now we need to bound the discrepancy between local and global errors $\parallel F(\theta_Q) - F_n(\theta_Q) \parallel$. Consider function $h(t) = \theta_0 + t(\theta_Q - \theta_0)$, then it holds that

$$F(\theta_Q) - F(\theta_0) = \int_0^1 \nabla F(h(t))^T(\theta_Q - \theta_0)dt \tag{38}$$

$$F_n(\theta_Q) - F_n(\theta_0) = \int_0^1 \nabla F_n(h(t))^T(\theta_Q - \theta_0)dt \tag{39}$$

Subtract the two equations and take the norm to get

$$\parallel F_n(\theta_Q) - F(\theta_Q) \parallel \leq \parallel F_n(\theta_0) - F(\theta_0) \parallel + \sigma_2 \parallel \theta_Q - \theta_0 \parallel \tag{40}$$

Then based on (21), (37) and (40), we can describe the score estimation error as

$$\sum_{k=0}^{K-1} \gamma_k \mathbb{E}\parallel s_{\theta_Q}(Y_{n,t_k}, T-t_k) - \nabla \log q(Y_{n,t_k}) \parallel^2$$

$$\leq \sum_{k=0}^{K-1} \gamma_k \mathbb{E}\parallel s_{\theta_Q}(Y_{n,t_k}, T-t_k) - \nabla \log q(Y_{n,t_k}|X_{n,0}) \parallel^2 + \sum_{k=0}^{K-1} \gamma_k C$$

$$= F_n(\theta_Q) + C(T - \delta)$$

$$\leq F(\theta_Q) + \parallel F_n(\theta_Q) - F(\theta_Q) \parallel + C(T - \delta)$$

$$= \mathcal{O}\Big(F(\theta_0) + \big(\frac{\sqrt{\Gamma^* S Q} L^2 w^4}{3Q(1-2w^2)} + \frac{3\sqrt{\Gamma^* S Q} L^2 w^4}{Q(1-2w^2)}\big)\mathbb{E}\parallel \theta_0 \parallel^2 + \frac{3LN\sigma_1^2}{2\Gamma^*} + \frac{5\sqrt{\Gamma^* S Q}}{18}(\sigma_1^2 + \sigma_2^2)$$

$$+ \parallel F_n(\theta_0) - F(\theta_0) \parallel + \sigma_2 \parallel \theta_Q - \theta_0 \parallel + C(T - \delta)\Big) \tag{41}$$

And according to the Theorem 1 in the work Benton et al. (2024), the Corollary 2 holds.

## G EXPERIMENTS

### G.1 EXPERIMENTAL SETUP

We conduct experiments using the Cifar-10 (Krizhevsky et al., 2009) SVHN (Netzer et al., 2011), and Fashion-MNIST (Xiao et al., 2017) datasets. To simulate a distributed learning scenario, we partition the training data among 10 workers. As described in Section 3.1, DDPM (Ho et al., 2020) can be viewed as a special case of our work, so we consider its distributed version (known as FedDM (Vora et al., 2024)) under resource-constrained conditions. In the experiments, we mainly consider two pruning techniques: Random Pruning (R) and Top-k Pruning (T) based on model weight. In particular, in order to explore the heterogeneity of pruning policy caused by resource differences among workers, we set for different pruning levels named F (Full), L (Large), M (Medium) and S (Small):

- **F:** All workers with full model;
- **L:** 80% workers with full model, and 20% workers with 75% model parameters;
- **M:** 60% workers with full model, 20% workers with 80% model parameters, and 20% workers with 75% model parameters;
- **S:** 60% workers with full model, and 40% workers with 75% model parameters.

We utilize multiple metrics to evaluate the performance of distributed training diffusion models with different pruning levels: Training loss is used to assess the convergence for distributed learning of score estimation. Additionally, the Inception Score (IS) and Fréchet Inception Distance (FID) are employed to evaluate the quality of data generation.

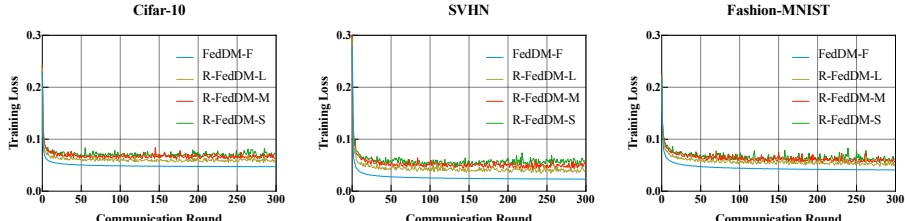

Figure 1: Training loss of FedDM under the random pruning with different pruning levels

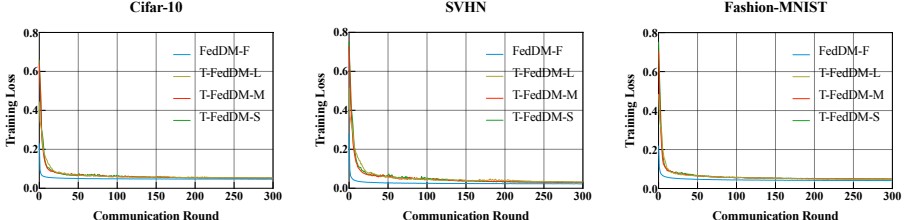

Figure 2: Training loss of FedDM under the Top-k pruning with different pruning levels

In the training stage of obtaining a score estimation, we use the U-Net backbone containing residual blocks (Tun et al., 2023). And we use the following settings unless otherwise stated: The number of communication rounds $Q$ is set as 300, the local training steps $S$ are configured as 5 epochs for Cifar-10 and 2 epochs for both SVHN and Fashion-MNIST, and the step size $\eta$ is 0.0001.

All the experiments s are implemented in PyTorch 2.5.1, Python 3.12, Cuda 12.1. And we run them on a Cloud Server with Intel(R) Xeon(R) Platinum 8358P CPU and total 10 RTX 3090 GPUs in Ubuntu 22.04.

## G.2 MODEL CONVERGENCE FOR DISTRIBUTED LEARNING OF SCORE ESTIMATION

We assess the convergence for distributed learning of score estimation on the above three datasets, using Random (R) and Top-k (T) pruning techniques. Specifically, we establish four pruning levels (F, L, M, and S) to observe the effects on convergence behavior. This series of experiments is designed to systematically evaluate how various levels of model sparsity influence the training dynamics.

Figures 1 and 2 illustrate the impact of different pruning strategies and pruning levels on the convergence rate of the distributed training diffusion model across three datasets. Overall, the training loss in all settings is effectively reduced as the number of communication rounds increases, verifying the effectiveness of the coordinate-wise aggregation method. Under both pruning strategies, as the degree of pruning increases (denoted by F, L, M, S), the training loss requires more communication rounds to decrease effectively, and the total reduction diminishes. This is because the reduced model introduces additional errors, which slows the convergence rate to a certain extent.

## G.3 DATA GENERATION QUALITY

We assess the performance of distributed training DDPM (known as FedDM) with different pruning levels on the above three datasets. Specifically, we establish four pruning levels (F, L, M, and S) and utilize two indicators, IS and FID, to observe and compare the average data generation quality.

As shown in Table 2, the experimental results demonstrate that pruning significantly impacts the performance of diffusion models in distributed learning, with the effects closely related to the pruning strategy, dataset complexity, and model heterogeneity. On complex datasets such as CIFAR-10 and SVHN, the full model (FedDM-F) achieves the best performance, while increased pruning levels lead to a substantial decline in the quality of random pruning (R-FedDM), as indicated by decreased IS scores and increased FID values, particularly at high pruning levels (e.g., S). In contrast, Top-k pruning (T-FedDM) better preserves model performance by retaining critical parameters, resulting in smaller increases in FID and performance closer to the full model, especially at moderate pruning levels (e.g., M). For simpler datasets like Fashion-MNIST, where the data distribution is less

Table 2: IS and FID comparison of FedDM with different pruning levels.

| Method | Cifar-10 | | SVHN | | Fashion-MNIST | |
|---|---|---|---|---|---|---|
| | IS ($\uparrow$) | FID ($\downarrow$) | IS ($\uparrow$) | FID ($\downarrow$) | IS ($\uparrow$) | FID ($\downarrow$) |
| FedDM-F | $4.59 \pm 0.13$ | 73.73 | $2.79 \pm 0.04$ | 163.36 | $3.58 \pm 0.08$ | 87.59 |
| R-FedDM-L | $3.95 \pm 0.12$ | 103.59 | $2.76 \pm 0.04$ | 93.78 | $3.47 \pm 0.04$ | 53.70 |
| R-FedDM-M | $4.01 \pm 0.08$ | 104.53 | $2.60 \pm 0.04$ | 127.47 | $3.32 \pm 0.08$ | 52.31 |
| R-FedDM-S | $3.60 \pm 0.07$ | 111.21 | $2.53 \pm 0.05$ | 120.57 | $3.46 \pm 0.07$ | 49.94 |
| T-FedDM-L | $4.39 \pm 0.08$ | 83.75 | $2.72 \pm 0.04$ | 157.19 | $3.59 \pm 0.07$ | 87.85 |
| T-FedDM-M | $4.54 \pm 0.10$ | 80.42 | $2.55 \pm 0.05$ | 146.27 | $3.54 \pm 0.06$ | 100.69 |
| T-FedDM-S | $4.31 \pm 0.13$ | 84.98 | $2.51 \pm 0.06$ | 193.84 | $3.63 \pm 0.07$ | 109.83 |

complex, pruning has a relatively smaller impact, and the performance difference between random pruning and Top-k pruning is minimal. Additionally, on Fashion-MNIST, higher pruning levels unexpectedly improve FID values. This phenomenon can be attributed to the lower capacity requirements of simple data distributions, where high pruning reduces redundant parameters, acting as a regularization effect to prevent overfitting, thus smoothing the generated distribution and making it closer to the real distribution. Model heterogeneity introduced by pruning is another critical factor affecting global performance, with random pruning more likely to cause aggregation errors, while Top-k pruning alleviates this issue to some extent. Overall, Top-k pruning proves more advantageous for complex datasets, while random pruning is better suited for resource-constrained scenarios involving simpler tasks. Future work can focus on optimizing pruning strategies and aggregation algorithms to further balance model efficiency and performance across various data distributions and task requirements.

## H  SOME ADDITIONAL DISCUSSION

**Relaxed Assumptions and Improved Convergence Result:** In deriving the convergence rate for training the score estimation model in a distributed manner, our proof builds on the work of Zhou et al., with the following key differences: 1) We eliminate their reliance on the bounded gradient assumption by modeling the iteration relationship. 2) By carefully handling the pruning error, we achieve the ultimate goal of gradient descent-based methods, allowing the final average gradient norm to converge to a little constant. Compared to their result, which converges only to a scaled version of $\frac{1}{Q} \sum_{q=1}^{Q} \mathbb{E} \parallel \theta_q \parallel^2$, our approach transforms the uncertain dependency in the convergence result into a deterministic one. 3) We achieve a convergence rate of $\mathcal{O}(\frac{1}{\sqrt{\Gamma^* SQ}})$ by adjusting parameters such as the step size $\eta$, improving upon their result of $\mathcal{O}(\frac{1}{\sqrt{Q}})$.

**Error Bound Refinement and Controllable Convergence:** Directly using our analytical framework to integrate the theoretical results of Zhou et al. (2024) (Theorem 1 in their paper) with the single-worker diffusion model generation error bound, we obtain the following error bound:

$$\text{KL}(q_{n,\delta} \parallel p_{n,t_K}) = \mathcal{O}\big(F(\theta_0) + \frac{3LN(\sigma_1^2 + \sigma_2^2)}{2S(\Gamma)^2} + \frac{L^2 NG}{2\Gamma\sqrt{Q}} + \frac{3L^2 w^2 N\sqrt{Q}}{\Gamma^*} \cdot \frac{1}{Q} \sum_{q=1}^{Q} \mathbb{E} \parallel \theta_q \parallel^2 +$$

$$\parallel F_n(\theta_0) - F(\theta_0) \parallel + \sigma_2 \parallel \theta_Q - \theta_0 \parallel + C(T - \delta) + \kappa dT + \kappa^2 dK + de^{-2T}\big)$$

The above error bound includes an uncertainty term $\frac{1}{Q} \sum_{q=1}^{Q} \mathbb{E} \parallel \theta_q \parallel^2$, which prevents the bound from being tightened by adjusting $Q$. This limitation restricts their ability to improve the error bound in collaborative training. In contrast, our approach eliminates this uncertainty by leveraging the model iteration relationship, transforming it into a deterministic dependency. We also show that the error bound can be effectively tightened by adjusting $Q$, as discussed in our Remark 1. This offers a clear advantage over their results.

**Unified Analytical Framework to Bridge Diffusion Models and Distributed Learning:** We propose a novel framework that bridges these two areas of diffusion models and distributed learning, providing the first unified approach to connect their theoretical foundations. Specifically, we propose a simple yet effective analytical approach based on function construction (Lines 460-465) to bridge the theoretical error bounds between distributed diffusion model training and single-worker diffusion model training. Notably, this analytical approach is applicable to any generation error bound obtained under the assumption on perfect score approximation in a single-worker paradigm. We chose to integrate with the work of Benton et al. (2024) as their results represent the current state-of-the-art results in a single-worker paradigm. In fact, as long as the theoretical generation error bound in the single-worker mode based on the perfect score assumption is developed into a better result, our analytical framework allows for an immediate extension to the corresponding distributed training error bound.

**Limitations and Future Work:** There are still some limitations in our work, which inspire some future research directions. As discussed in Remark 1, smaller $w^2$ helps tighten the bound on $\mathcal{O}(\epsilon^2)$, which limits the level of pruning. Therefore, in practice, how to directly strike a balance between resource consumption and error tolerance is still worth exploring. Therefore, it is necessary to design a suitable pruning strategy according to the specific task to balance model performance and resource consumption. Additionally, resource constraints are only considered when training the score estimation model during the reverse process. However, noise schedule during the forward process may still encounter similar constraints, which we will leave for the future.

