# OpenReview forum: "$d$-Linear Generation Error Bound for Distributed Diffusion Models"
_ICLR.cc/2025/Conference — Submitted to ICLR 2025_

### Official Review · Reviewer_wu8x · 2024-11-01

**Soundness:** 2
**Presentation:** 3
**Contribution:** 1
**Rating:** 3
**Confidence:** 3

**Summary:**

This paper studied the distributed learning problem on diffusion models, where each agent performs multiple local updates on a sparsified diffusion model using sparsified gradients. A convergence analysis is provided.

**Strengths:**

- The proof sketch is well-presented.

**Weaknesses:**

- For clarity, please denote equation numbers for important equations such as those in the Theorem, Corollary and Proof Sketch.
- Also for clarity, I suggest denoting $B_1, B_2, B_3, B_4, B_5$ with their corresponding iteration counter, e.g., $B_{1}^{(q)}$.
- (**Experiment**) This paper lacks a numerical experiment to show the effectiveness of distributed diffusion model training using only sparse local updates, which will help identify the contribution of the proposed theorem.
- (**Theoretical Contribution**) I agree with the authors that the relaxed assumption of this paper improves the theoretical results of [Benton et al., 2024]. However, since the convergence of federated local sparse training is already established in [Zhou et al, 2024] and is orthogonal to the diffusional model aspect, the combination of two techniques makes the contribution of this paper a bit ambiguous.
- (**Low Practical Contribution**) Unless the authors can point out additional algorithmic design on training diffusion models using (7) as a contribution beyond the classification task as illustrated in [Zhou et al, 2024], the analysis in this paper will not impact the community in advancing distributed diffusion model training.

**Questions:**

- Using the upper bound $B_3 = \mathcal{O}(\eta^2 \sum_{p=0}^{q-1} \\| \theta_{p+1} - \theta_p \\|^2 + \eta^2 \\| \theta_0 \\|^2 + \eta^4 + \eta^4 \\| \nabla F(\theta_q) \\|^2 )$ from line 428, summing $B_3$ from $q= 0$ to $q= Q-1$ and applying the inequalities from line 350, 357 will give us
$$\frac{1}{Q}\sum_{q=0}^{Q-1} \mathbb{E}[ \\| \nabla F(\theta_q) \\|^2 ] = \tilde{\mathcal{O}}(\mathbb{E}\\| \theta_0 \\|^2)$$
which is different from the $\tilde{\mathcal{O}}(\frac{1}{\eta^2 Q^2}\mathbb{E}\\| \theta_0 \\|^2)$ dependence as claimed in Theorem 1. (I use the notation $\tilde{\mathcal{O}}$ to hide the dependence on other error terms.)
For instance, may I ask for clarification on how does the $\mathcal{O}(\eta Q \mathbb{E}[\\| \theta_0 \\|^2])$ dependence in line 857 reduce to $\mathcal{O}(\frac{1}{\eta Q}\mathbb{E}[\\| \theta_0 \\|^2])$ in line 863.

---

> ### Author Response · Authors · 2024-11-22
> **Response to Weakness1, Weakness2 and Weakness3**
>
> We thank the reviewer wu8x for the time and valuable feedback! We would try our best to address the comments one by one.
>
> **Response to Weakness1 \& Weakness2**
>
> Thank you for your constructive suggestions. We agree that adding equation numbers to important equations enhance clarity. We update the manuscript to include these equation numbers for ease of reference.
>
> Regarding the notation of $B_1$, $B_2$, $B_3$, $B_4$, $B_5$, your suggestion to include the iteration counter, is well-taken. We revise the notation accordingly to explicitly reflect the iteration counters, which improve the readability and consistency of the presentation.
>
> **Response to Weakness3**
>
> Thank you for your comments. We understand that such an experiment would provide direct evidence of the practical impact of our proposed theorem.
> In response, we conduct a numerical experiment that demonstrates the effectiveness of our approach in Appendix G.
>
> Specifically, we implement distributed diffusion model training incorporating sparse local updates as described in the paper. Figures 1 and 2 illustrate the impact of different pruning strategies and pruning levels on the convergence rate of the distributed training diffusion model across three datasets.
> Table 2 demonstrates that pruning significantly impacts the performance of diffusion models in distributed learning, with the effects closely related to the pruning strategy, dataset complexity, and model heterogeneity.

---

> ### Author Response · Authors · 2024-11-22
> **Response to Weakness4**
>
> **Response to Weakness4**
>
> Thank you for your insightful feedback regarding the theoretical contribution of our paper.
> We appreciate your recognition that the relaxed assumption improves upon the results of [Benton et al., 2024].
> To better highlight the distinctions between our work and prior studies, we provide the following detailed comparisons:
>
> **Comparison with [Zhou et al., 2024]:**
>
> **1. Relaxed Assumptions and Improved Convergence Results:**
>
> 1) [Zhou et al., 2024] relies on the strong bounded gradient assumption (Assumption 3 in their paper), which limits its applicability in practical scenarios.
>
> 2) Their theoretical results only show that the average gradient norm converges to a constant value proportional to $\frac{1}{Q}\sum_{q=1}^Q \mathbb{E}\parallel \theta_q \parallel^2$. However, ours can converge to 0, thereby meeting the theoretical goal of gradient-based optimization methods.
>
> 3) We achieve a faster convergence rate of $\mathcal{O}(\frac{1}{\sqrt{\Gamma^* SQ}})$ by adjusting parameters such as the step size $\eta$, improving upon their result of $\mathcal{O}(\frac{1}{\sqrt{Q}})$. This advancement underscores the critical roles of the number of local training steps $S$ and the minimum parallel training degree $\Gamma^*$ in enhancing convergence efficiency.
>
> **2. Error Bound Refinement and Controllable Convergence:**
> If using our analytical framework to directly integrate the theoretical results of Zhou et al. (Theorem 1 in their paper) with the single-worker diffusion model generation error bound, the following error bound would emerge:
> $$
> \begin{aligned}
> KL(q_{n,\delta}\parallel p_{n,t_K})
> =&\mathcal{O}\big( F(\theta_0)+
> \frac{3LN(\sigma_1^2+\sigma_2^2)}{2S(\Gamma^*)^2}+\frac{L^2 N G}{2\Gamma^*\sqrt{Q}}+\frac{3L^2 w^2 N \sqrt{Q}}{\Gamma^*}\cdot\frac{1}{Q}\sum_{q=1}^Q\mathbb{E}\parallel \theta_q \parallel^2\notag\\\\
> &+\parallel F_n(\theta_0)-F(\theta_0) \parallel+\sigma_2\parallel \theta_Q-\theta_0 \parallel+C(T-\delta) +\kappa d T+\kappa^2 d K+d e^{-2T}
> \big)\notag
> \end{aligned}
> $$
> Due to incomplete analysis of pruning errors, the error bound in [Zhou et al., 2024] includes an uncertainty term $\frac{1}{Q}\sum_{q=1}^Q \mathbb{E}\parallel \theta_q \parallel^2$. Our work eliminates this uncertainty by leveraging the model iteration relationship, transforming it into a deterministic dependency. We also demonstrate that the error bound can be effectively **tightened** by adjusting $Q$, as discussed in Remark 1. This improvement offers a clear advantage over their results.
>
> **Comparison with [Benton et al., 2024]:**
>
> **1. Extension to Distributed Framework:**
> [Benton et al., 2024] focuses on theoretical error bounds for single-worker diffusion models and does not address the influence of training dynamic.
> **We reveal the explicit impact of the complications of distributed training** (such as the number of training rounds $Q$, the number of local training steps $S$ between two communication rounds, and the number of workers $N$) **on the final generation error bound, by analyzing the true iteration loss.**
>
> **2. Unified Analytical Framework to Bridge Diffusion Models and Federated Learning:**
> We propose a novel framework that bridges the two areas of diffusion models and federated learning, providing the first unified approach to connect their theoretical foundations. Specifically, we propose a simple yet effective analytical approach based on function construction (Lines 460-465) to bridge the theoretical error bounds between distributed diffusion model training and single-worker diffusion model training. Notably, this analytical approach is applicable to any generation error bound obtained under the assumption on perfect score approximation in a single-worker paradigm.
>
> We sincerely hope that you can re-evaluate the contribution of our theoretical work.

---

> ### Author Response · Authors · 2024-11-22
> **Response to Weakness5 and Question**
>
> **Response to Weakness5**
>
> We respectfully disagree with the assessment that our work lacks practical contribution. **The primary contribution of this paper lies in its theoretical advancements, which are foundational for distributed diffusion model training.** Specifically, we provide **the first rigorous generation error bound** for distributed diffusion models. This work goes beyond algorithmic design and addresses a fundamental gap in the theoretical understanding of distributed training diffusion models.
>
> We emphasize that advancing theory is essential for ensuring the robustness and reliability of distributed training diffusion models. Without such a theoretical framework, the design and deployment of practical algorithms risk being ad hoc and unprincipled.
>
> We sincerely hope that you can re-evaluate the contribution of our theoretical work.
>
> **Response to Question**
>
> Of course, we can provide further clarification of the derivation process from Line 857 to Line 863. The key to this inequality lies in the precise control of the step size $\eta$, which ensures that the desired bounds are achieved. Specifically, Lines 824-830 provide critical support for this result by establishing the conditions under which the step size remains effective. To make this explanation more concrete, we take the term $\frac{2\eta S L^2NQw^2}{\Gamma^*(1-8\eta^2L^2S^2)}\mathbb{E} \parallel \theta_0\parallel^2$ in the inequality as an example and explain it in detail.
>
> As detailed in Lines 824-830, if the step size $\eta$ satisfies the condition $\eta \le \sqrt{\frac{\Gamma^*}{48S^2L^2Q^2N+8S^2L^2\Gamma^*}}$, it holds that
> $$\frac{12\eta^2S^2L^2Q^2w^2N}{\Gamma^*(1-8\eta^2L^2S^2)}\le \frac{w^2}{4}.$$
>
> Using these results, we can derive:
> $$
> \begin{aligned}
> \frac{2\eta S L^2NQw^2}{\Gamma^*(1-8\eta^2L^2S^2)}\mathbb{E} \parallel \theta_0\parallel^2=&\frac{12\eta^2S^2L^2Q^2w^2N}{\Gamma^*(1-8\eta^2L^2S^2)} \cdot \frac{2\eta S L^2NQw^2}{12\eta^2S^2L^2Q^2w^2N}\mathbb{E} \parallel \theta_0\parallel^2\notag\\\\
> \le &\frac{w^2}{4}\cdot \frac{1}{6 \eta S Q}\mathbb{E} \parallel \theta_0\parallel^2= \frac{w^2}{24 \eta S Q}\mathbb{E} \parallel \theta_0\parallel^2\notag
> \end{aligned}
> $$
> The derivations for the other terms in the inequality follow similarly, using analogous steps and the same reasoning framework.
>
> If there are any further questions, we are happy to clarify and try to address them. Thank you again and your recognition means a lot for our work.

---

> > ### Comment · Reviewer_wu8x · 2024-11-22
> > **Response on the step size condition**
> >
> > Thank you the authors for the response. The step size condition $\eta \leq \sqrt{\frac{\Gamma^*}{48S^2 L^2 Q^2 N + 8 S^2 L^2 \Gamma^*}} = \Theta(\frac{1}{SQ})$ seems to contradicts with the step size choice $\eta = \Theta(\sqrt{\frac{\Gamma^*}{SQ}})$ used in Corollary 1. For instance, when $SQ$ is large, $\eta$ must be as small as $\Theta(\frac{1}{SQ})$. May I ask the authors to further explain on the choice of $\eta$ in Corollary 1, i.e., its dependence w.r.t. $Q$ and $S$?

---

> > > ### Author Response · Authors · 2024-11-22
> > > **Response on the step size condition**
> > >
> > > Thank you for your comments. We appreciate your attention to detail and hope to address the concern about whether there is a contradiction between the two step-size conditions.
> > >
> > > **Our response is that the two step size conditions are not contradictory.** The specific proof is as follows:
> > >
> > > For simplicity, we mark $\eta \le \sqrt{\frac{\Gamma^*}{48S^2L^2Q^2N+8S^2L^2\Gamma^*}}$ as condition 1 and $\eta=\Theta(\sqrt{\frac{\Gamma^*}{SQ}})$ as condition 2.
> > >
> > >
> > > Since the following relationship obviously holds:
> > >
> > > $\exists$ constant c , $ s.t.$ $\eta = c \sqrt{\frac{\Gamma^*}{SQ}}\quad \Rightarrow \quad \eta=\Theta(\sqrt{\frac{\Gamma^*}{SQ}})$
> > >
> > >
> > > Therefore, to prove that conditions 1 and 2 are not contradictory, we only need to prove there exists at least one $c$ making the following be satisfied:
> > >
> > > $$c \sqrt{\frac{\Gamma^*}{SQ}}\le \sqrt{\frac{\Gamma^*}{48S^2L^2Q^2N+8S^2L^2\Gamma^*}}   \qquad(1)$$
> > >
> > > Simplifying this inequality (1), we can get
> > >
> > > $$c \le \sqrt{\frac{1}{48S L^2Q N+8\frac{S}{Q}L^2\Gamma^*}} \qquad (2)$$
> > >
> > > Since $c \in \mathbb{R}$, we can find infinite $c$ that satisfy inequality (1).
> > >
> > > So we conclude that the two step size choices are not contradictory.
> > >
> > > Thanks again for your feedback and we hope this proof clarify your confusion. If you have any further questions, we would be happy to answer them.

---

> > > > ### Comment · Reviewer_wu8x · 2024-11-22
> > > > **Discussion of the step size condition**
> > > >
> > > > In the above $c$, it has a $\mathcal{O}(1/\sqrt{Q})$ dependence w.r.t. $Q$, therefore it is not constant w.r.t. $Q$. I remark that I assume the notation $\Theta(\cdot)$ only hides the constants that are **independent** w.r.t. $Q, S$ and $\Gamma^*$.

---

> ### Author Response · Authors · 2024-11-25
>
> **Thank you very much for pointing out this error!** Through careful derivation, we found that the reason for the contradiction is that we did not deduce the bound $B_5^{(q)}$ of the pruning error tightly enough, which led to **the mismatch of the order of the bound $B_4^{(q)}$ and $B_5^{(q)}$**. Therefore, **we revised the derivation of $B_5^{(q)}$ (Lines 800-858)**, and obtained the following Corollary 1:
>
> Under Assumptions 1-3, if the step size $\eta$ satisfies $\eta=\sqrt{\frac{\Gamma^*}{SQ}}$, and pruning factor satisfies $w \in [0, \frac{\sqrt{2}}{2})$, and we can further set $Q \ge \max\\{ 729\Gamma^* SL^2,\Gamma^* S, 3\Gamma^* ( 8SL^2+\frac{16SL^2 w^2}{1-2w^2} )  \\}$ to further derive the convergence result of Theorem 1:
> $$
> \begin{aligned}
> \frac{1}{Q} \sum_{q=0}^{Q-1}\mathbb{E} \parallel\nabla F(\theta_{q})\parallel^2
> \le\frac{6(F(\theta_0)-F(\theta_Q))}{\sqrt{\Gamma^* SQ}}+(\frac{2L^2w^4}{Q(1-2w^2)}+\frac{18L^2 w^4}{Q(1-2w^2)})\mathbb{E} \parallel  \theta_{0}\parallel^2+\frac{9L N \sigma_1^2}{\Gamma^* \sqrt{\Gamma^* SQ}}+\frac{5}{3}(\sigma_1^2+\sigma_2^2)\notag
> \end{aligned}
> $$
> In this corollary, we make **the average gradient norm converge to a small constant at a rate of $\mathcal{O}(\frac{1}{\sqrt{\Gamma^{*} SQ}})$**, and further reveal the low tolerance of diffusion model training to errors, which is caused by the sampling randomness and the potential inconsistency of the objective function, **as discussed in our Remark 1 and Conclusion**.
>
> (The rest of the relevant parts have been modified)
>
> Thank you again for your great help in correcting it!

---

### Official Review · Reviewer_qExk · 2024-11-02

**Soundness:** 2
**Presentation:** 2
**Contribution:** 2
**Rating:** 3
**Confidence:** 3

**Summary:**

This paper presents a theoretical framework for distributed diffusion models, specifically focusing on deriving a theoretical error bound for model generation in resource-constrained settings. The authors extend the analysis of score estimation convergence under arbitrary pruning and introduce a theoretical generation error bound that scales linearly with data dimension $d$. The contributions include assessing the convergence of the score estimation model and error propagation in a distributed system with arbitrary pruning.

**Strengths:**

1. The paper's theoretical analysis highlights the performance of distributed diffusion models under resource limitations such as arbitrary pruning, which is valuable for real-world applications.

2. The derived error bounds align with existing single-worker setup, demonstrating the validity and potential applicability of the proposed framework in distributed settings.

3. It aligns with recent advances in federated diffusion and distributed learning by considering score estimation dynamics, bridging gaps in understanding error bounds across distributed workers.

**Weaknesses:**

1. **Incremental Contribution:** While the theoretical contributions in the distrusted diffusion models are noteworthy, the paper's essence is to solve a federated learning problem. This approach appears to be incremental when compared to prior works such as Zhou et al. 2024, Benton et al. 2024. See questions below for more details.

2. **No Empirical Validation:** The paper lacks empirical results that validate the theoretical error bound derived for distributed diffusion models. Experiments demonstrating the practical impact of pruning and convergence rates in resource-constrained environments would strengthen the claims.

3. **Clarity and Accessibility:** The proof in this paper is not self-contained as it lacks some explanations and leaves the gap to the readers.

**Questions:**

1. While the paper includes some comparisons with prior works in the proofs, it does not clearly differentiate its contributions or demonstrate substantial advancements beyond what was previously established. Can the authors provide a clear explanation of the fundamental technical novelty and methodology, especially in comparison to works such as [Zhou et al., 2024; Benton et al., 2024]? Specifically, it would be helpful to understand what unique aspects this work introduces in terms of theoretical framework or analysis techniques that set it apart from these prior results.

2. Could the authors elaborate on how summing (24) around L.811 leads to (25)?

3. The proof of Corollary 2 in Appendix E is not self-contained because the authors rely largely on Theorem 1 in the work [Benton et al. 2024]. Would you please provide a step-by-step explanation of how to adapt Theorem 1 from Benton et al. 2024 to fit this distributed setting, and highlight any required modifications or additional assumptions?

---

> ### Author Response · Authors · 2024-11-22
> **Response to Weakness1, Question1, and Weakness2**
>
> We thank the reviewer qExk for the time and valuable feedback! We would try our best to address the comments one by one.
>
> **Response to Weakness1 \& Question1**
>
> Thank you for your constructive comments. To better highlight the distinctions between our work and prior studies, we provide the following detailed comparisons:
>
> **Comparison with [Zhou et al., 2024]:**
>
> **1. Relaxed Assumptions and Improved Convergence Results:**
>
> 1) [Zhou et al., 2024] relies on the strong bounded gradient assumption (Assumption 3 in their paper), which limits its applicability in practical scenarios.
>
> 2) Their theoretical results only show that the average gradient norm converges to a constant value proportional to $\frac{1}{Q}\sum_{q=1}^Q \mathbb{E}\parallel \theta_q \parallel^2$. However, ours can converge to 0, thereby meeting the theoretical goal of gradient-based optimization methods.
>
> 3) We achieve a faster convergence rate of $\mathcal{O}(\frac{1}{\sqrt{\Gamma^* SQ}})$ by adjusting parameters such as the step size $\eta$, improving upon their result of $\mathcal{O}(\frac{1}{\sqrt{Q}})$. This advancement underscores the critical roles of the number of local training steps $S$ and the minimum parallel training degree $\Gamma^*$ in enhancing convergence efficiency.
>
> **2. Error Bound Refinement and Controllable Convergence:**
> If using our analytical framework to directly integrate the theoretical results of Zhou et al. (Theorem 1 in their paper) with the single-worker diffusion model generation error bound, the following error bound would emerge:
> $$
> \begin{aligned}
> KL(q _ {n,\delta}\parallel p _ {n,t _ K})
> =& \mathcal{O}( F(\theta _ 0) +\frac{3LN(\sigma _ 1^2+\sigma _ 2^2)}{2S(\Gamma^*)^2} + \frac{L^2 N G}{2\Gamma^* \sqrt{Q}} + \frac{3L^2 w^2 N \sqrt{Q}}{\Gamma^*} \cdot \frac{1}{Q}\sum _ {q=1}^Q\mathbb{E}\parallel \theta _ q \parallel^2\\\\ &+ \parallel F _ n(\theta _ 0)-F(\theta _ 0) \parallel+ \sigma _ 2\parallel \theta _ Q-\theta _ 0 \parallel + C(T-\delta) + \kappa d T + \kappa^2 d K + d e^{-2T} )
> \end{aligned}
> $$
>
> Due to incomplete analysis of pruning errors, the error bound in [Zhou et al., 2024] includes an uncertainty term $\frac{1}{Q}\sum_{q=1}^Q \mathbb{E}\parallel \theta_q \parallel^2$. Our work eliminates this uncertainty by leveraging the model iteration relationship, transforming it into a deterministic dependency. We also demonstrate that the error bound can be effectively **tightened** by adjusting $Q$, as discussed in Remark 1. This improvement offers a clear advantage over their results.
>
> **Comparison with [Benton et al., 2024]:**
>
> **1. Extension to Distributed Framework:**
> [Benton et al., 2024] focuses on theoretical error bounds for single-worker diffusion models and does not address the influence of training dynamic. **We reveal the explicit impact of the complications of distributed training** (such as the number of training rounds $Q$, the number of local training steps $S$ between two communication rounds, and the number of workers $N$) **on the final generation error bound, by analyzing the true iteration loss.**
>
> **2. Unified Analytical Framework to Bridge Diffusion Models and Federated Learning:**
> We propose a novel framework that bridges the two areas of diffusion models and federated learning, providing the first unified approach to connect their theoretical foundations. Specifically, we propose a simple yet effective analytical approach based on function construction (Lines 460-465) to bridge the theoretical error bounds between distributed diffusion model training and single-worker diffusion model training. Notably, this analytical approach is applicable to any generation error bound obtained under the assumption on perfect score approximation in a single-worker paradigm.
>
> We sincerely hope that you can re-evaluate the contribution of our theoretical work.
>
> **Response to Weakness2**
>
> Thank you for your constructive comments to improve our work. As you wish, we add an experimental section in Appendix G of the submitted manuscript to show the effects of different degrees of pruning on the convergence rate and generation quality of the diffusion model trained in a distributed manner.
>
> In Appendix G, Figures 1 and 2 illustrate the effects of various pruning strategies and levels on the convergence rate of the distributed training diffusion model across three datasets. Table 2 further highlights the significant impact of pruning on the performance of diffusion models in distributed learning, showing that the effects are closely tied to the pruning strategy, dataset complexity, and model heterogeneity.

---

> > ### Comment · Reviewer_qExk · 2024-11-24
> > **Response to Authors' comparisons with previous works**
> >
> > Thank the authors for the clarification of their works and I acknowledge the revision in the updated manuscript. However, I have concerns about the author's claim about their bound converging to zero with rate $O(\\frac{1}{\\sqrt{\\Gamma^* SQ}})$ by choosing $\\eta = \\Theta(\\sqrt{\\frac{\\Gamma^*}{SQ}})$. As pointed out by reviewer wu8x,
> > > this choice of $\\eta$ contradicts the step size condition $\eta \leq \sqrt{\frac{\Gamma^*}{48S^2L^2Q^2N+8S^2L^2\Gamma^*}}$ when $SQ$ is large enough.
> >
> > and I agree with reviewer wu8x's opinion on why $\\eta = \\Theta(\\sqrt{\\frac{\\Gamma^*}{SQ}})$ shouldn't be the case. From my perspective, under the step size condition in Theorem 1, there seems no choice of $\\eta$ to make $\\eta SQ \\to 0$ in the average gradient bound as $SQ \\to \\infty$.

---

> ### Author Response · Authors · 2024-11-22
> **Response to Weakness3 and Question3**
>
> **Response to Weakness3 \& Question3**
>
> Thanks for your comments. To address your questions about Appendix E (currently updated to Appendix F due to revisions), we hope the following explanations will clarify any confusion.
>
> **Problem Setting:** In our distributed setting, the inverse process of the diffusion model, i.e., the process of recovering the original image distribution from the noise distribution, is performed independently by each worker. This inverse process is consistent with the current process of training diffusion models with a single worker, such as the work of Benton et al.. Our framework differs from the current single-worker paradigms mainly in whether the forward process trains the score estimation neural network in a distributed or a single-worker manner. In the currently popular single-worker paradigm, the training complexity of the score estimation neural network model has been ignored. For example, the generation error results of Benton et al. depend on a constant $\epsilon_{\text{score}}^2$, which defined by $\sum\limits_{k=0}^{K-1} \gamma_k \mathbb{E}\parallel s_{\theta_Q}(Y_{n,t_k},T-t_k)-\nabla \log q(Y_{n,t_k})\parallel^2 \le \epsilon_{\text{score}}^2$. Our goal is to explore which factors theoretically affect $\sum\limits_{k=0}^{K-1} \gamma_k \mathbb{E}\parallel s_{\theta_Q}(Y_{n,t_k},T-t_k)-\nabla \log q(Y_{n,t_k})\parallel^2$, rather than simply assuming a constant error bound $\epsilon_{\text{score}}^2$.
>
> Next, we provide a step-by-step explanation of how to adapt Theorem 1 from Benton et al. to this distributed context.
>
> **Step 1. Analyze the local errors caused by denoising score matching.** In the score estimation model training, the term $\sum\limits_{k=0}^{K-1} \gamma_k \mathbb{E}\parallel s_{\theta}(Y_{n,t_k},T-t_k)-\nabla \log q(Y_{n,t_k})\parallel^2$ cannot be directly utilized as the local loss due to the inaccessibility of $\nabla \log q(\cdot)$, which depends on the unknown data distribution.  Instead, the denoising score matching technique is employed to redefine the local loss as $F_n(\theta)=\sum_{k=0}^{K-1} \gamma_k \mathbb{E}_{X_{n,0}, q(Y_{n,t_k}|X_{n,0})}[\parallel s_{\theta}(Y_{n,t_k},T-t_k)- \nabla \log q(Y_{n,t_k}|X_{n,0})\parallel^2]$ (Our Eq. (9)). This reformulation leverages the conditional distribution $q(Y_{n,t_k}|X_{n,0})$ to circumvent the direct dependency on $\nabla \log q(\cdot)$, making it practical for implementation. However, the resulting score estimation error introduced by this substitution must be carefully analyzed, as discussed in the lines 452-460 of our paper.
>
> **Step 2. Analyze the discrepancy between the global loss $F(\theta_Q)$ and the local loss $F_n(\theta_Q)$.** Our Corollary 1 reveals the relationship between the average gradient norm and the global loss $F(\theta_{Q})$. By rearranging terms in the inequality, we derive an upper bound for $F(\theta_{Q})$, which is presented in Lines 437-443. Since the previous step of analyzing the denoising score matching error requires the definition of $F_n(\theta_Q)$, we aim to obtain it using the inequality $F_n(\theta_Q) \leq \parallel F_n(\theta_Q)-F(\theta_{Q})\parallel + F(\theta_{Q})$. The key challenge, therefore, lies in bounding $\parallel F_n(\theta_Q)-F(\theta_{Q})\parallel$, which we address through the construction of a specially designed auxiliary function. The details of this construction are provided in Lines 460-469.

---

> ### Author Response · Authors · 2024-11-22
> **Response to Question2**
>
> **Response to Question2**
>
> Of course, we can explain the derivation process of Eq. (24) to (25) in detail for you.
>
> Eq. (24):
> $$
> \begin{aligned}
> B _ 2
> =&\frac{L}{2}\mathbb{E}\parallel \theta_{q+1}-\theta_{q}\parallel^2\notag\\\\
> \le&\frac{3NLS\eta^2\sigma_1^2}{2(\Gamma^*)^2}+\frac{3LS^2 \eta^2}{2}\mathbb{E} \parallel\nabla F(\theta_{q})\parallel^2+\frac{3L}{2} B_3\notag\\\\
> \le&\frac{3NLS\eta^2\sigma_1^2}{2(\Gamma^*)^2}+\frac{3LS^2 \eta^2}{2}\mathbb{E} \parallel\nabla F(\theta_{q})\parallel^2+\frac{6\eta^2S^2L^3qw^2N}{\Gamma^*(1-8\eta^2L^2S^2)}\sum_{p=0}^{q-1}\mathbb{E} \parallel \theta_{p+1}-\theta_p\parallel^2+\notag\\\\
> &\frac{6\eta^2S^2L^3w^2N}{\Gamma^*(1-8\eta^2L^2S^2)}\mathbb{E} \parallel \theta_0\parallel^2+\frac{12\eta^4S^4L^3N(\sigma_1^2+\sigma_2^2)}{\Gamma^*(1-8\eta^2L^2S^2)}+\frac{12\eta^4S^4L^3N}{\Gamma^*(1-8\eta^2L^2S^2)}\mathbb{E} \parallel \nabla F(\theta_q)\parallel^2 \qquad(24)
> \end{aligned}
> $$
>
> Summing it from $q=0$ to $Q-1$:
> $$
> \begin{aligned}
> &\frac{L}{2}\sum_{q=0}^{Q-1}\mathbb{E}\parallel \theta_{q+1}-\theta_{q}\parallel^2\notag\\\\
> \le&\frac{3\eta^2SLQN\sigma_1^2}{2(\Gamma^*)^2}+\frac{3\eta^2 S^2 L}{2}\sum_{q=0}^{Q-1}\mathbb{E} \parallel\nabla F(\theta_{q})\parallel^2+\frac{6\eta^2S^2L^3Q w^2N}{\Gamma^*(1-8\eta^2L^2S^2)}\sum_{q=0}^{Q-1}\sum_{p=0}^{q}\mathbb{E} \parallel \theta_{p+1}-\theta_p\parallel^2+\notag\\\\
> &\frac{6\eta^2S^2L^3 Q w^2N}{\Gamma^*(1-8\eta^2L^2S^2)}\mathbb{E} \parallel \theta_0\parallel^2+\frac{12\eta^4S^4L^3 Q N(\sigma_1^2+\sigma_2^2)}{\Gamma^*(1-8\eta^2L^2S^2)}+\frac{12\eta^4S^4L^3N}{\Gamma^*(1-8\eta^2L^2S^2)}\sum_{q=0}^{Q-1}\mathbb{E} \parallel \nabla F(\theta_q)\parallel^2\notag\\\\
> \le&\frac{3\eta^2SLQN\sigma_1^2}{2(\Gamma^*)^2}+\frac{3\eta^2 S^2 L}{2}\sum_{q=0}^{Q-1}\mathbb{E} \parallel\nabla F(\theta_{q})\parallel^2+\frac{12\eta^2S^2L^2 Q^2 w^2N}{\Gamma^*(1-8\eta^2L^2S^2)}\cdot \frac{L}{2}\sum_{q=0}^{Q-1}\mathbb{E} \parallel \theta_{q+1}-\theta_q\parallel^2+\notag\\\\
> &\frac{6\eta^2S^2L^3 Q w^2N}{\Gamma^*(1-8\eta^2L^2S^2)}\mathbb{E} \parallel \theta_0\parallel^2+\frac{12\eta^4S^4L^3 Q N(\sigma_1^2+\sigma_2^2)}{\Gamma^*(1-8\eta^2L^2S^2)}+\frac{12\eta^4S^4L^3N}{\Gamma^*(1-8\eta^2L^2S^2)}\sum_{q=0}^{Q-1}\mathbb{E} \parallel \nabla F(\theta_q)\parallel^2\notag
> \end{aligned}
> $$
>
> Then the following (25) can be derived by rearranging terms:
> $$
> \begin{aligned}
> &[1-\frac{12\eta^2S^2L^2Q^2w^2N}{\Gamma^*(1-8\eta^2L^2S^2)}]\cdot \frac{L}{2}\sum_{q=0}^{Q-1}\mathbb{E}\parallel \theta_{q+1}-\theta_{q}\parallel^2\notag\\\\
> \le&\frac{3\eta^2SLQN\sigma_1^2}{2(\Gamma^*)^2}+\frac{6\eta^2S^2L^3QNw^2}{\Gamma^*(1-8\eta^2L^2S^2)}\mathbb{E} \parallel \theta_0\parallel^2+\frac{12\eta^4S^4L^3QN(\sigma_1^2+\sigma_2^2)}{\Gamma^*(1-8\eta^2L^2S^2)}+\notag\\\\
> &[\frac{3 \eta^2S^2L}{2}+\frac{12\eta^4S^4L^3N}{\Gamma^*(1-8\eta^2L^2S^2)}]\sum_{q=0}^{Q-1}\mathbb{E} \parallel \nabla F(\theta_q)\parallel^2\qquad (25)
> \end{aligned}
> $$
>
> If there are any further questions, we are happy to clarify and try to address them. Thank you again and your recognition means a lot for our work.

---

> > ### Comment · Reviewer_qExk · 2024-11-24
> > **Response to the detailed derivation process of Eq. (24) to (25)**
> >
> > Thank the author for the explanation. However, I still don't quite understand the second inequality in the step of summing from $q=0$ to $Q-1$. Specifically, I'm looking at the part
> >
> > $$
> > \frac{L}{2}\sum_{q=0}^{Q-1}\mathbb{E}\parallel \theta_{q+1}-\theta_{q}\parallel^2 \leq [some~~ terms] +\frac{6\eta^2S^2L^3Q w^2N}{\Gamma^*(1-8\eta^2L^2S^2)}\sum_{q=0}^{Q-1}\sum_{p=0}^{q}\mathbb{E} \parallel \theta_{p+1}-\theta_p\parallel^2 \leq [some~~ terms]+\frac{12\eta^2S^2L^2 Q^2 w^2N}{\Gamma^*(1-8\eta^2L^2S^2)}\cdot \frac{L}{2}\sum_{q=0}^{Q-1}\mathbb{E} \parallel \theta_{q+1}-\theta_q\parallel^2
> > $$
> >
> > I know that the pre-factor $\frac{6\eta^2S^2L^3Q w^2N}{\Gamma^*(1-8\eta^2L^2S^2)} = \frac{12\eta^2S^2L^2 Q^2 w^2N}{\Gamma^*(1-8\eta^2L^2S^2)}\cdot \frac{L}{2}$, but my confusion stems from the missing summation $\sum_{p=0}^{q}\mathbb{E} \parallel \theta_{p+1}-\theta_p\parallel^2$ in the second inequality. From your expression, how come $\sum_{p=0}^{q}\mathbb{E} \parallel \theta_{p+1}-\theta_p\parallel^2 \leq \mathbb{E} \parallel \theta_{q+1}-\theta_q\parallel^2$, can the authors make more comments on this?
> >
> > Since one contribution in this paper is the fine-grained analysis on $\\mathbb{E}[\\|\\theta_{q+1} - \\theta_q \\|^2]$ that was missing in [Zhou et al., 2024], I would like the authors to further clarify this step.

---

> > > ### Comment · Reviewer_qExk · 2024-11-24
> > > **Response to the detailed derivation process of Eq. (24) to (25)**
> > >
> > > I see what the authors mean. There's a missing step
> > > $$\sum_{q=0}^{Q-1}\sum_{p=0}^q\\mathbb{E} \parallel\theta_{p+1} - \theta_p \parallel^2 =  \sum_{p=0}^{Q-1}\sum_{q=p}^{Q-1}\\mathbb{E} \parallel\theta_{p+1} - \theta_p \parallel^2 = \sum_{p=0}^{Q-1}(Q-p) \\mathbb{E} \parallel\theta_{p+1} - \theta_p \parallel^2  \leq Q \sum_{p=0}^{Q-1}\\mathbb{E} \parallel\theta_{p+1} - \theta_p \parallel^2.$$
> > >
> > > I have no problem with this derivation and would suggest the authors to add detailed explanation for this derivation.

---

> ### Author Response · Authors · 2024-11-25
>
> Thanks for pointing out this inconsistency! We have responded to Reviewer wu8x in detail regarding this issue：
>
> Through careful derivation, we found that the reason for the contradiction is that we did not deduce the bound $B_5^{(q)}$ of the pruning error tightly enough, which led to **the mismatch of the order of the bound $B_4^{(q)}$ and $B_5^{(q)}$**. Therefore, **we revised the derivation of $B_5^{(q)}$ (Lines 800-858)**, and obtained the following Corollary 1:
>
> Under Assumptions 1-3, if the step size $\eta$ satisfies $\eta=\sqrt{\frac{\Gamma^*}{SQ}}$, and pruning factor satisfies $w \in [0, \frac{\sqrt{2}}{2})$, and we can further set $Q \ge \max\\{ 729\Gamma^* SL^2,\Gamma^* S, 3\Gamma^* ( 8SL^2+\frac{16SL^2 w^2}{1-2w^2} )  \\}$ to further derive the convergence result of Theorem 1:
> $$
> \begin{aligned}
> \frac{1}{Q} \sum_{q=0}^{Q-1}\mathbb{E} \parallel\nabla F(\theta_{q})\parallel^2
> \le\frac{6(F(\theta_0)-F(\theta_Q))}{\sqrt{\Gamma^* SQ}}+(\frac{2L^2w^4}{Q(1-2w^2)}+\frac{18L^2 w^4}{Q(1-2w^2)})\mathbb{E} \parallel  \theta_{0}\parallel^2+\frac{9L N \sigma_1^2}{\Gamma^* \sqrt{\Gamma^* SQ}}+\frac{5}{3}(\sigma_1^2+\sigma_2^2)\notag
> \end{aligned}
> $$
> In this corollary, we make **the average gradient norm converge to a small constant at a rate of $\mathcal{O}(\frac{1}{\sqrt{\Gamma^{*} SQ}})$**, and further reveal the low tolerance of diffusion model training to errors, which is caused by the sampling randomness and the potential inconsistency of the objective function, **as discussed in our Remark 1 and Conclusion**.

---

### Official Review · Reviewer_LfB7 · 2024-11-09

**Soundness:** 3
**Presentation:** 1
**Contribution:** 3
**Rating:** 5
**Confidence:** 3

**Summary:**

This paper studies the iteration complexity of sampling from a diffusion model in the distributed setting. It shows an $O(d)$ bound, which matches the performance in the single-worker setting. The main contribution is a careful analysis of the discrepancy between the distributed optimization of the score matching objective, and the analogous optimization in the serial setting, and the effect of this discrepancy on the final sampling error. It essentially shows that using Q = poly(1/$\varepsilon$) rounds of distributed training suffice to achieve this bound, where $\varepsilon$ is the parameter controlling the final sampling TV error.

**Strengths:**

This is a very interesting paper that proposes a new setting for theoretical analysis of diffusion models. The bound shown matches the best known bound in the serial worker setting. Moreover, I believe that this paper could open the door to other analyses in the distributed setting that could lead to practical impact. Overall, I appreciate the creativity here.

**Weaknesses:**

Despite the nice technical contributions, this paper was extremely painful to read. The theorem statements and corollaries are very difficult to parse, and the writing in general is convoluted and unclear. I encourage the authors to consider rewriting the main text of the paper, and to focus on explaining the core intuition crisply. I also recommend significantly simplifying the theorem statements.

**Questions:**

- Can you get a bound better than O(d) if you make the assumption that the score is Lipschitz?
- Does your work have any direct practical implications?
- What open problems does your work expose?

---

> ### Author Response · Authors · 2024-11-22
>
> We thank the reviewer LfB7 for the time and valuable feedback! We would try our best to address the comments one by one.
>
> **Response to Weakness**
>
> Thank you for your comments. We have made the following improvements:
>
> 1. We reorganize the main text of the paper, with the main changes involving Sections 4 and 5.
>
> 2. Considering that this paper involves numerous notations and formulas, we add a notation table in the Table 1 of Appendix A to help readers better interpret the content.
>
> 3. We provide more detailed explanations of the key theorems and corollaries in Section 4-5 and Appendix H to ensure that the core results are clearer and easier to understand.
>
> **Response to Question1**
>
> We hope our response clarifies your doubts regarding our avoidance of assuming the score is Lipschitz.
>
> **In fact, the uniform Lipschitz assumption is restrictive for some cases.** For example, the Lipschitz constant may implicitly scale with the data dimension, which can occur when the data distribution is approximately supported on a sub-manifold. Furthermore, using a time-varying Lipschitz constant will not automatically resolve these issues. As an illustration, Chen et al. (2023) assume Lipschitz smoothness at $t=0$ but still obtain a quadratic dependence on the data dimension $d$. Based on these reasons, we choose to avoid the Lipschitz assumption on the score.
>
> **Response to Question2**
>
> In response to your concern about the direct practical implications of our work,
>
> 1. We believe it provides theoretical insights for distributed diffusion model training under practical resource constraints. For example, diffusion models can help restore image details lost in low-light or high-noise environments. Since equipments such as cameras are typically decentralized and limited in storage and computing power, pruning techniques are a reasonable choice for implementing distributed diffusion model training.
>
> 2. The error bound (Corollary 1) we provide offers a theoretical assessment of image restoration quality for the above scenario, while the discussion in Remark 1 gives guidance on setting key parameters.
>
> 3. To further illustrate the impact of pruning on distributed training of diffusion models, we add an experimental section in Appendix G that explores the effects of different pruning levels on the training of popular diffusion models in a distributed setting.
>
> **Response to Question3**
>
> Thank you for your feedback on the improvement of our work. We have added a discussion of the open problems revealed by this work in
> Appendix H.
>
> To our knowledge, we are the first to establish convergence and error bounds for distributed training diffusion models.
> However, there are still some limitations in our work, which inspire some future research directions. As discussed in Remark 1, larger $\Gamma^*$ and smaller $w^2$ help tighten the bound on $\mathcal{O}(\epsilon^2)$. However, finding the optimal balance between a low $w^2$ and a high $\Gamma^*$ in model extraction remains a challenging task. Additionally, resource constraints are only considered when training the score estimation model during the reverse process. However, noise schedule during the forward process may still encounter similar constraints.
>
> If there are any further questions, we are happy to clarify and try to address them. Thank you again and your recognition means a lot for our work.

---

> > ### Author Response · Authors · 2024-11-26
> > **Further revisions to Question3**
> >
> > Due to the comments of other reviewers, we revised the discussion of limitations in Appendix H:
> >
> > **Limitations and Future Work:** There are still some limitations in our work, which inspire some future research directions. As discussed in Remark 1, smaller $w^2$ helps tighten the bound on $\mathcal{O}(\epsilon^2)$, which limits the level of pruning. Therefore, it is necessary to design a suitable pruning strategy according to the specific task to balance model performance and resource consumption. Additionally, resource constraints are only considered when training the score estimation model during the reverse process. However, noise schedule during the forward process may still encounter similar constraints, which we will leave for the future.
> >
> > Thank you once again!

---

> > > ### Comment · Reviewer_LfB7 · 2024-11-28
> > >
> > > Thank you for your responses. I still feel that the presentation is lacking, and can be significantly improved. I would recommend that the authors significantly rewrite the main paper, and focus on providing a clean exposition of their contributions, especially relative to (Zhou et al, 2024). As someone unfamiliar with this work, it is very difficult for me to understand what your contribution is relative to this work from reading your paper. Moreover, the theorem statements are still very difficult to understand without significant effort. I would have liked to see a crisp explanation of exactly what it is you are trying to accomplish, and your core contributions.
> > >
> > > I don't think this paper is ready for publication in the current state. For these reasons, I maintain my score.

---

### Meta-Review · Area_Chair_CKn8 · 2024-12-10

**Metareview:**

This paper applies distributed training techniques to the score estimation, which combined with existing convergence results of diffusion models, leads to an error bounds for training distributed diffusion models. The analysis in this paper largely borrows from the separate literature in distributed training and diffusion models, and therefore fails short in terms of novelty.

**Additional Comments On Reviewer Discussion:**

While the authors were able to clarify some points during the rebuttal such as hyperparameter choices, the main caveat of this paper still stands.

---

### Decision · Program_Chairs · 2025-01-22

Reject